# Visual Anagrams Reveal Hidden Differences in Holistic Shape Processing Across Vision Models

**Fenil R. Doshi**
Dept. of Psychology
& Kempner Institute
Harvard University
fenil_doshi@fas.harvard.edu

**Thomas Fel**
Kempner Institute
Harvard University
tfel@g.harvard.edu

**Talia Konkle**
Dept. of Psychology
& Kempner Institute
Harvard University
talia_konkle@harvard.edu

**George A. Alvarez**
Dept. of Psychology
& Kempner Institute
Harvard University
alvarez@wjh.harvard.edu

## Abstract

Humans are able to recognize objects based on both local texture cues and the configuration of object parts, yet contemporary vision models primarily harvest local texture cues, yielding brittle, non-compositional features. Work on shape-vs-texture bias has pitted shape and texture representations in opposition, measuring shape relative to texture, ignoring the possibility that models (and humans) can simultaneously rely on both types of cues, and obscuring the absolute quality of both types of representation. We therefore recast shape evaluation as a matter of absolute configural competence, operationalized by the **Configural Shape Score (CSS)**, which *(i)* measures the ability to recognize both images in *Object-Anagram pairs* that preserve local texture while permuting global part arrangement to depict different object categories. Across 86 convolutional, transformer, and hybrid models, CSS *(ii)* uncovers a broad spectrum of configural sensitivity with fully self-supervised and language-aligned transformers – exemplified by DINOv2, SigLIP2 and EVA-CLIP – occupying the top end of the CSS spectrum. Mechanistic probes reveal that *(iii)* high-CSS networks depend on long-range interactions: radius-controlled attention masks abolish performance showing a distinctive U-shaped integration profile, and representational-similarity analyses expose a mid-depth transition from local to global coding. A BagNet control, whose receptive fields straddle patch seams, remains at chance *(iv)*, ruling out any "border-hacking" strategies. Finally, *(v)* we show that configural shape score also predicts other shape-dependent evals (e.g.,foreground bias, spectral and noise robustness). Overall, we propose that the path toward truly robust, generalizable, and human-like vision systems may not lie in forcing an artificial choice between shape and texture, but rather in architectural and learning frameworks that seamlessly integrate both local-texture and global configural shape. [1]

## 1 Introduction

Human object recognition is remarkably robust: we can effortlessly identify objects across dramatic variations in texture, scale, viewpoint, and context because we can focus on aspects of global

---

[1]Project Page: https://www.fenildoshi.com/configural-shape/

39th Conference on Neural Information Processing Systems (NeurIPS 2025).

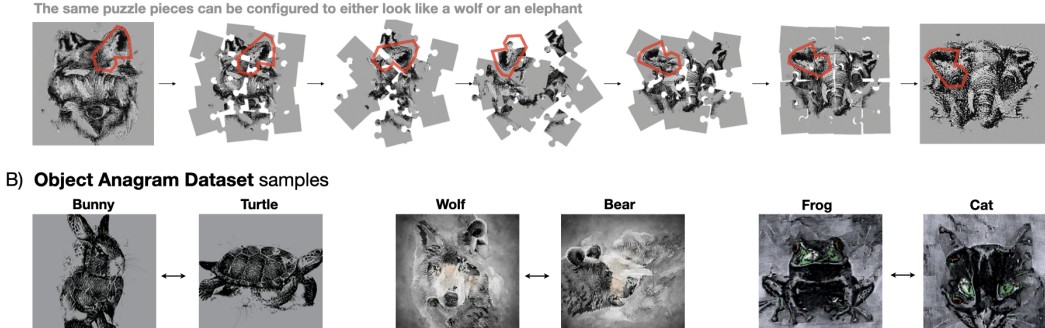

A) **Visual Anagram** example
The same puzzle pieces can be configured to either look like a wolf or an elephant

B) **Object Anagram Dataset** samples

Bunny  Turtle  Wolf  Bear  Frog  Cat

Figure 1: **Object-Anagram task: a probe of configural shape perception. (A)** visual-anagram example—an identical set of 16 square diffusion patches is spatially permuted to form two distinct objects, here a wolf and an elephant (one shared patch is outlined in red). **(B)** additional image pairs from the object-anagram benchmark. each pair comprises globally different objects built from the same unordered patch multiset, forcing any successful classifier to rely solely on the global arrangement of parts.

configuration that are stable across such local photometric quirks [1, 2, 3, 4, 5, 6]. By contrast, state-of-the-art vision networks still harvest local, high-frequency shortcuts [7, 8, 9]. This strategy achieves high ImageNet accuracy [10] but leaves models brittle under texture shifts, adversarial noise, and compositional out-of-distribution stresses [11, 12, 13, 14]. The failure arises because models often seize on spurious yet linearly separable features when multiple predictive cues are available [15, 16, 17]. These differences between models and humans are often studied using the shape–versus–texture bias diagnostic, which pits shape vs. texture using cue-conflict stimuli [8], but this metric is inherently relative: scores rise whenever shape coding strengthens *or* when texture coding weakens, rendering the absolute fidelity of global shape ambiguous [18, 19, 20]. Effective vision systems should exploit both cues when helpful [21, 22, 23], motivating an absolute assessment of shape and texture processing A.1.

We close this gap by recasting shape evaluation as an absolute test of configural competence. Building on "visual anagrams" [24], we synthesize image–pairs that share an identical multiset of local diffusion patches yet differ in their permutations (Fig. 1). Correctly classifying both views demands sensitivity to spatial relations alone. We formalise the task through the **Configural Shape Score (CSS)**, a joint two–image criterion whose chance level is below $2\%$ and whose ceiling mandates perfect configural sensitivity. Our study benchmarks 86 convolutional, transformer, and hybrid checkpoints— from BagNet [25], stylized [8] and adversarially robust CNNs [26], to fully self-supervised ViTs like DINOv2 [27, 28], and language-aligned models such as SigLIP [29, 30] and EVA-CLIP [31]. Combining behavioral metrics with mechanistic probes yields five key findings:

- The nine-category *Object-Anagram* dataset enforces a stringent falsification test for holistic vision by permuting the global arrangement of an invariant multiset of local patches; any success therefore hinges on configural integration. The Configural Shape Score over this dataset gives an absolute score of configural shape, rigorously decoupling genuine shape inference from the artefactual gains that cue-conflict paradigms can achieve through mere texture suppression.

- Vision transformers optimized via self-supervised learning and language-alignment, notably DINOv2 [27], EVA-CLIP [31] and SigLIP2 [30], dominate the CSS spectrum; their global-consistency objectives appear uniquely effective at instilling holistic shape representations, whereas comparably accurate, purely supervised counterparts achieve lower configural shape scores.

- Mechanistic dissection reveals that high-CSS models leverage cross-patch communication spanning long-range interactions: performance collapses under radius-clipped attention masks with a *U-shaped* integration profile indicating that intermediate layers perform the key configural processing; representational-similarity analyses echo this profile, exposing a mid-depth pivot from local to global coding that is predicitive of overall CSS score.

- Architectures confined to local receptive fields, exemplified by BagNet, perform near chance, ruling out "border-hacking" and underscoring that authentic configural shape demands long-range integration.
- Models with higher configural shape scores also score high on other shape-dependent evals like foregorund-vs-background bias, robustness to noise, phase dependence and critical band masking.

By converting a long-standing theoretical critique into a falsifiable measurement and linking the resulting scores to identifiable computational mechanisms, this work advances the science of holistic shape perception and offers actionable design principles for future vision systems.

## 2 Related Work

**Configural and Holistic Shape Processing in Human Vision:** In humans, there is evidence that configural shape processing is multifaceted, and the term broadly encapsulates any computation where the precise arrangement of parts affects the representation of object appearance or identity [1, 4, 5]. There is even evidence that the appearance and recognition of local parts can be influenced by long-range interactions with other distal parts [32], indicating that contextual modulation is an important component of configural processing. These forms of configural shape processing can be distinguished from texture-based representations, where items can appear to have the same texture despite spatial shifts in local features or parts, as long as the key higher-order statistical properties are the same [33, 34].

**Computational Approaches to shape sensitivity:** Prior work has investigated shape representations and texture bias in vision models [8, 7, 35, 36, 37, 38, 39], often framing these issues in terms of shortcut learning driven by spurious correlations in the training data [15, 13]. Building on this, more recent studies have begun to probe whether models are sensitive to the spatial configuration of object parts [40, 41, 42, 43]. However, these efforts typically rely on synthetic datasets and/or require explicit fine-tuning, focusing on understanding whether an architecture is at all capable of supporting relational reasoning, rather than whether such sensitivities emerge naturally during training. A notable exception is work by Baker and Elder [9], who tested whether pretrained models could detect disruptions in object configuration. Their approach involved splitting silhouette images along the horizontal meridian, flipping the bottom half, and stitching the parts back together. This manipulation was intended to break global structure while preserving local part content. However, this manipulation has some limitations: it is ineffective for symmetric shapes, and the use of black-and-white silhouettes can obscure subtle configural differences.

## 3 Object Anagram Dataset and Configural Shape Score (CSS)

**Background and notation.** Consider the classical supervised-learning paradigm, where $\mathcal{X}$ denotes the image space and $\mathcal{Y} = [C]$ the index set of $C$ distinct categories. A classifier $\boldsymbol{f} : \mathcal{X} \to \mathcal{Y}$ maps an input image $\boldsymbol{x}$ to a predicted label $\boldsymbol{y}$. Our goal is to probe configural shape acuity: the ability to parse global part arrangements while remaining invariant to permutations of local texture elements. To do so we fix a grid of $K = 16$ equal-area square patches. Let $\mathfrak{S}_K$ be the symmetric group of the $K!$ possible permutations, and write $\boldsymbol{\pi} \in \mathfrak{S}_K$ for an element thereof. Given an ordered multiset of patches $\mathcal{P} = \{\boldsymbol{p}_k\}_{k=1}^K$ with $\boldsymbol{p}_k \in \mathbb{R}^{h \times w \times 3}$, we define the composition operator:

$$\boldsymbol{\Gamma}(\mathcal{P}, \boldsymbol{\pi}) = \begin{bmatrix} \boldsymbol{p}_{\pi(1)} & \cdots & \boldsymbol{p}_{\pi(4)} \\ \vdots & \ddots & \vdots \\ \boldsymbol{p}_{\pi(13)} & \cdots & \boldsymbol{p}_{\pi(16)} \end{bmatrix} \in \mathcal{X},$$

which re-assembles the permuted patches into a $256 \times 256$ canvas. The permutation $\boldsymbol{\pi}$ therefore fully determines the global layout.

**Object Anagram Dataset synthesis.** The synthesis pipeline is directly adapted from [24]. For every ordered label pair $(y_1, y_2) \in \mathcal{Y}^2$ we prepare a text–layout tuple $\big(\boldsymbol{c}(y_j), \boldsymbol{\pi}_j\big)_{j=1,2}$, where $\boldsymbol{c}(y_j)$ encodes the prompt *"high-quality painting of a well-shown $y_j$ with simple black paint texture on a grey background"* using a pretrained T5 encoder, and where $\boldsymbol{\pi}_1 = \mathrm{id}$ while $\boldsymbol{\pi}_2 \neq \boldsymbol{\pi}_1$ is drawn uniformly from $\mathfrak{S}_K$. Both tuples share a common Gaussian seed, ensuring identical low-level texture statistics, and are injected into the DeepFloyd-IF pipeline[2]. To maintain texture consistency while

---

[2]Available at : `https://github.com/deep-floyd/IF`.

supporting distinct global configurations, we use a permutation operator $\Pi(\boldsymbol{z}, \boldsymbol{\pi})$ that rearranges $\boldsymbol{z}$ according to $\boldsymbol{\pi}$. At each reverse-diffusion timestep $t$, we compute two denoising predictions: $\boldsymbol{\epsilon}^{(1)}$ for the canonical arrangement $(y_1)$ and $\boldsymbol{\epsilon}^{(2)}$ for the permuted arrangement $(y_2)$. Formally:

$$\boldsymbol{\epsilon}^{(1)} = \boldsymbol{\varepsilon_\theta}(\boldsymbol{z}_t, t, \boldsymbol{c}(y_1)), \quad \boldsymbol{\epsilon}^{(2)} = \boldsymbol{\varepsilon_\theta}(\Pi(\boldsymbol{z}_t, \boldsymbol{\pi}_2), t, \boldsymbol{c}(y_1)).$$

These are combined into a symmetrized target

$$\boldsymbol{\epsilon}_t = \boldsymbol{\epsilon}^{(1)} + \Pi^{-1}(\boldsymbol{\epsilon}^{(2)}, \boldsymbol{\pi}_2),$$

where $\Pi^{-1}(\cdot, \boldsymbol{\pi})$ inverses the permutation so that the two predictions align in the canonical frame. The reverse-diffusion update is then

$$\boldsymbol{z}_{t-1} = \frac{1}{\sqrt{\alpha_t}}\Big(\boldsymbol{z}_t - \frac{1-\alpha_t}{\sqrt{1-\bar{\alpha}_t}}\,\boldsymbol{\epsilon}_t\Big) + \sigma_t\,\boldsymbol{\eta}_t, \quad \boldsymbol{\eta}_t \sim \mathcal{N}(\boldsymbol{0}, \boldsymbol{I}), \qquad \bar{\alpha}_t = \prod_{s=1}^{t} \alpha_s,$$

with a cosine noise schedule $\alpha_t$ and variance $\sigma_t^2 = 1 - \alpha_t$. This procedure jointly optimizes both category representations, using identical patch content but differing spatial arrangements, progressively refining a shared image. In each timestep, we obtain $\boldsymbol{z}_0$ at $64 \times 64$ resolution and after $T$ steps the resulting image seeds a second diffusion at $256 \times 256$ resolution. From the final image we extract the patch multiset $\mathcal{P}$ by partitioning it into a $4 \times 4$ grid, yielding sixteen patches that share texture but differ in arrangement across the two views:

$$\boldsymbol{x}^{(1)} = \boldsymbol{\Gamma}(\mathcal{P}, \boldsymbol{\pi}_1), \qquad \boldsymbol{x}^{(2)} = \boldsymbol{\Gamma}(\mathcal{P}, \boldsymbol{\pi}_2),$$

with ground-truth labels $(y^{(1)}, y^{(2)})$. The critical property of these image pairs is texture invariance: $(\boldsymbol{x}^{(1)}, \boldsymbol{x}^{(2)})$ share exactly the same patch multiset, and therefore identical first- and many higher-order texture statistics (color distributions, edge patterns, local frequencies) while differing solely in global configuration. Consequently, local cues alone are insufficient for classification, making this dataset a stringent test of a model's configural processing capabilities.

**Configural Shape Score.** Gathering $N$ such pairs yields $\mathcal{A} = \{(\boldsymbol{x}_i^{(1)}, \boldsymbol{x}_i^{(2)}, y_i^{(1)}, y_i^{(2)})\}_{i=1}^{N}$. Each image is centre-cropped to $224 \times 224$, normalised by the training statistics of $\boldsymbol{f}$, and forwarded through the network. Mapping the resulting ImageNet logits to the nine object-anagram categories (Appendix A.3) we define

$$\mathrm{CSS}(\boldsymbol{f}) = \frac{1}{N} \sum_{i=1}^{N} \mathbb{1}\big(\boldsymbol{f}(\boldsymbol{x}_i^{(1)}) = y_i^{(1)} \wedge \boldsymbol{f}(\boldsymbol{x}_i^{(2)}) = y_i^{(2)}\big),$$

whose chance level is $1/C^2$. Suppressing texture alone cannot raise the score; only a genuinely holistic integration of global layout yields high CSS values.

## 4 Vision Models

To dissect the computational determinants of configural shape sensitivity we assembled a suite of 86 pretrained models and four randomly initialized baselines that together span the principal axes of modern visual representation learning. Standard convolutional networks trained with cross-entropy on ImageNet: ResNet-50[44], VGG-16[45], and AlexNet [46] establish a supervised point of comparison, while three targeted manipulations of this template probe whether amplifying shape-vs-texture bias alone suffices: Stylized models [8], Adversarially Robust models [26], and Top-k Sparse models [47]. Architecturally bio-inspired models, including the CORnet family[48, 49], Long-Range Modulatory CNNs [50], and an Edge-AlexNet trained exclusively on edge statistics, test the hypothesis that neural plausibility intrinsically fosters holistic processing. The role of sheer data exposure is examined through the BiT checkpoints [51, 52] together with SWSL-ResNet-50 and SSL-ResNet-50 trained on billion-image corpora [53]. Recent architectural refinements are covered by ConvNeXt architectures [54, 55] as well as by ResNet-50, ResNet-101 and ViT-B/16 checkpoints trained with rigorous augmentation pipelines [56, 57]. A second axis contrasts convolutional and transformer principles. To this end, we include supervised Vision Transformers[58] along with its self-supervised counterparts: BEiT and BEiTv2 ViTs [59, 60], MAE-ViTs and Hiera-MAE-ViTs [61, 62], and DINOv2-ViTs [27, 28]. We also add version with language aligned encoders such as

CLIP ViTs [63, 64], SigLIP ViTs and SigLIP2 ViTs [29, 30], and EVA-CLIP ViTs[31, 65]. Within each ViT family, we included several variants (e.g. S/M/L variations). These comparisons isolate the contribution of transformer and multimodal objectives. Finally, BagNets [25], whose receptive fields never exceed local neighborhoods, serves as explicit local-only controls.

Collectively, this curated but diverse cohort will allow us to identify which architectural, training, and data regimes are predictive of high Configural Shape Score (see A.11 for the complete list of all the evaluated models)..

## 5 Models differ substantially in reliance on configural information for recognition

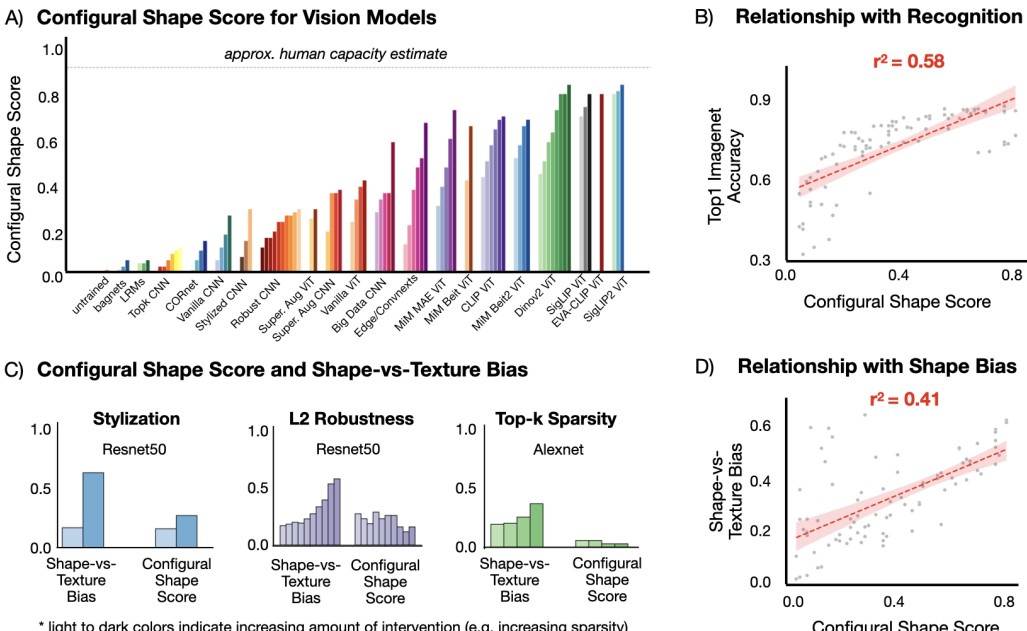

Figure 2: **Configural Shape Score (CSS) reveals variation across vision models matched in recognition performance and dissociates from imagenet accuracy and shape-vs-texture bias.** (A) CSS across 86 vision models, quantifying how accurately models recognize the distinct objects in each anagram pair. Human performance is shown as the dashed reference line. (B) Relationship between CSS and top-1 Imagenet Accuracy across all models. (C) CSS compared to shape-vs-texture bias for models trained with stylization, adversarial robustness, and Top-K sparsity. While these methods increase shape-vs-texture bias, they show modest-to-no gains in CSS. (D) Relationship between CSS and Shape-vs-Texture bias across all models.

Configural Shape Score varies widely across the full suite of models tested (Fig 2A), with the highest-scoring models approaching human-level scores (see A.4 for human experiment details), and the lowest scoring models demonstrating little-to-no configural shape sensitivity at all. Models that showed the highest CSS were either self-supervised ViTs(DINOv2s) or language-aligned ViTs (SigLIP and EVA-CLIP models). It is also notable that models with similar, generally high levels of ImageNet top-1 accuracy vary markedly in their CSS scores. For example, a supervised ViT-B/16 (ImageNet top-1: 76.35%; 23.61% CSS) and a language-aligned SigLIP ViT-B/16(ImageNet top-1: 74.96%; 77.78% CSS have near equivalent ImageNet top-1 accuracy, but achieve this through different reliance on configural shape information. Thus, achieving high accuracies on ImageNet is not enough to obtain a high Configural Shape Score, and ImageNet accuracy alone does not determine the Configural Shape Score (Fig. 2B).

Configural Shape Score also dissociates from Shape-vs-Texture bias. As shown in Fig. 2C, CSS is unaffected by three key strategies known to enhance shape-vs-texture bias: stylization-based training [8], adversarial training [26], and top-k activation pruning [47]. In stylization training, object textures are decorrelated from object identity during training, forcing models to rely more on shape than the

object's texture. Fig. 2C (left) shows that models trained with stylization (dark blue) have much higher shape-bias than models trained without stylization (light blue), but there's little to no effect of stylization on configural shape score. Adversarial training optimizes models for robustness to worst-case perturbations, varying strength of the adversarial attack with an epsilon parameter. Fig. 2C (center) shows that shape-bias increases with epsilon (purple), while CSS is unaffected by this manipulation. Finally, Top-k activation pruning restricts forward propagation to the highest-activating units within each layer, and increasing sparsity via this pruning (green bars) increases shape-bias but has no effect on configural shape score (2C right). Across these manipulations, we find that gains in CSS were modest-to-none compared to the substantial increases in shape bias. Finally, overall we find that the correlation between CSS and shape-vs-texture bias is moderate (r=0.64) indicating that only about 41% of the variance in CSS is accounted for by shape-vs-texture bias and vice versa (2D).

Taken together, these results indicate that the Configural Shape Score varies widely across models and dissociates from both ImageNet accuracy and Shape-vs-Texture bias. To gain deeper mechanistic insight into how models achieve high CSS, we next performed attentional ablation and representational similarity analyses.

# 6    Long-range Contextual Interactions lead to higher Configural Shape Scores in Vision Transformers

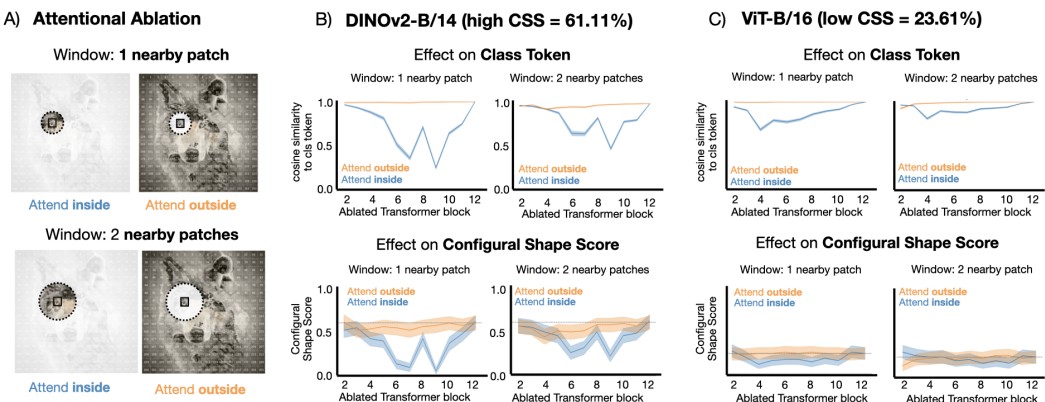

Figure 3: **Long-range Contextual Interactions leads to higher Configural Shape Score.** (A) Ablating self-attention in DINOv2-B/14 by selectively restricting each patch to attend only inside (blue) or outside (orange) a local window.Ablations are applied over windows with 1 or 2 nearby patches. (B) Effect of attentional ablation on the class token representation and configural shape score for high CSS model (Dinov2-B/14). Restricting attention to short-range interactions ("attend inside" condition - blue line) changes class tokens and disrupts CSS, most strongly at intermediate blocks. This effect is minimal when restricting attention to long-range interactions ("attend outside" condition - orange line). Dashed line shows CSS in unablated condition. (C) Effect of attentional ablation on the class token representation and configural shape score for low CSS model (ViT-B/16). Disruption for short-range interactions have reduced in this model.

Vision transformers provide a unique opportunity to examine the mechanisms of configural processing, because standard ViTs divide the image into a grid of patches and any configural processing (interactions between patch representations) must be performed via self-attention mechanisms. Thus, by targeting self-attention mechanisms with ablations, we can determine the relative impact of both short-range and long-range contextual interactions. Here we examined how intermediate attention mechanisms influence representational dynamics within DINOv2-B/14, a self-supervised ViT with 61.11% CSS and 84.1% top-1 ImageNet recognition. DINOv2-B/14 processes an input image of size 224×224 pixels by dividing it into a grid of 16×16 patches (each 14×14 pixels). For comparison, we also performed the ablation study on ViT-B/16, which achieved high top-1 ImageNet accuracy (76.35%) but had a low CSS (23.61%). We performed attentional ablations during inference at different intermediate stages of these models by selectively restricting each patch's attention within a targeted attention block.

To determine the relative impact of short-range and long-range attentional interactions, we defined two distinct attention masking conditions (Fig. 3A): (i) "attend inside," where each query patch attends only to patches within a specified Manhattan radius, and (ii) "attend outside," where attention is restricted to patches outside this radius. The class token was always allowed unrestricted attention in both conditions. We measured the cosine similarity of the original class token (unmasked) to the class token in both ablation conditions, as well as the CSS scores. If the class token representation depends mostly on short-range attention, then it should be unchanged for the "attend inside" masks, but should drop when "attending outside" (excluding critical short-range interactions). In contrast, if long-range interactions are most important, then the representation should be unchanged when attending outside, and should drop when attending inside (excluding the critical long-range interactions). We defined "short-range" attention interactions as those within a radius of 1 or 2 patches.

The results of these ablations for the high-CSS DINOv2 are shown in Fig. 3B. Attending only to very local neighborhoods (radius of 1 or 2 patches) substantially disrupted both class token representations and CSS (blue line, "attend inside" drops dramatically in middle layers). In contrast, attending only to more distant patches (orange curves, "attend outside") resulted in little-to-no change in class-token representations or CSS. Thus, it appears that attention interactions beyond at least 2 patches are necessary and sufficient to determine the class token representation and CSS score. Fig. 3C shows that a vision transformer (ViT-B/16) with lower CSS shows a reduced dependence on long-range interactions, suggesting that long-range attentional interactions are crucial for obtaining high-CSS.

Finally, these results show a U-shaped trend, suggesting that long-range interactions are particularly important in intermediate layers, and less important at early and late layers. These results suggest that early layers process patches relatively locally, then intermediate layers reinterpret and modify these local patch representations based on context, and then later stages aggregate locally over these contextually-modulated patch representations en route to the final model output. This observation is broadly consistent with work on LLMs, which suggest that early layers process text at the local token-level, followed by syntatic and broader contextual processing, and then finally return to more token-specific processing focusing on task-specific predictions and output generation [66, 67].

To further test the critical role of these intermediate layers, especially in deeper architectures where single-block ablations can be too subtle, we designed a more stringent cumulative ablation experiment on the full DINOv2 family. In this analysis, we measured the irrecoverable contribution of intermediate layers by systematically ablating long-range interactions in all blocks before or after a selected layer, and measuring it's impact on the class token's representation from the last block. The full methodology and results are detailed in A.5. The biggest drop was seen in the intermediate blocks across all model sizes, confirming that the mid-depth layers were indeed the locus of the most critical computations for configural processing in these models.

## 7 Relational Positional Encodings Boost Configural Shape Scores

While long-range attention is necessary for configural processing, the underlying mechanism for encoding spatial positions is also critical. To isolate this factor, we evaluated five variants of the supervised ViT-B/16 architecture, each with a different positional encoding (PE) scheme (Table 1). While the standard model using learned absolute PEs achieved a

Table 1: Configural Shape Scores (CSS) for ViT-B/16 variants.

| Model Variant (ViT-B/16) | CSS (%) |
| --- | --- |
| Relative Positional Embeddings | 38.89 |
| RoPE (Rotary Position Embedding) | 51.38 |
| RoPE (Rotary Position Embedding) + Absolute PE | 51.38 |
| Mixed RoPE | 48.61 |
| Mixed RoPE + Absolute PE | 52.77 |

CSS of 23.61%, models equipped with relative PEs, such as Rotary Position Embeddings (RoPE), more than doubled this score to over 51%. This result demonstrates that a model's ability to process the relational spatial structure of an image is critical for holistic shape perception, and that CSS rewards this capacity.

## 8 Disentangling the influence of object category and anagram puzzle pieces on configural shape score

The ablation experiments on vision transformers demonstrated the emergence of configural representations in the intermediate model layers. However, our ablation method only applies to models with

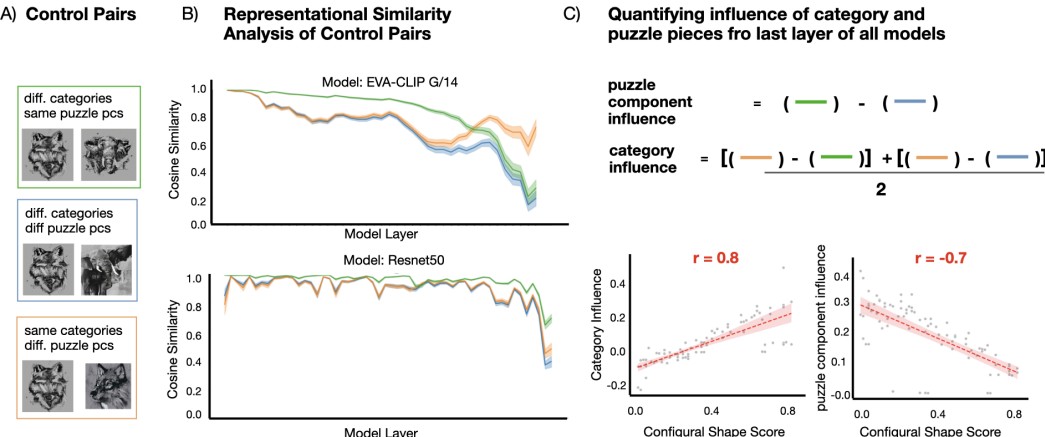

Figure 4: (A) Control pairs to tease apart category-level and component-level influence in model representations. (B) Cosine similarity across layers for each control pair type in EVA-CLIP G/14 and ResNet50. (C) Quantifying influence of object category vs. puzzle component from final layer embeddings. Models with higher Configural Shape Score (CSS) show stronger category influence and weaker component influence

self-attention mechanisms. To examine this transition from local to configural representations more generally for all model classes, we conducted a representational similarity analysis using a subset of carefully controlled image pairs from the Object Anagram Dataset. The purpose of this analysis was to determine at which point, if ever, different models transition from locally-driven representations dominated by puzzle-piece similarity to globally-driven representations dominated by categorical similarity. To disentangle the contributions of puzzle-piece similarity and configural-shape similarity, we measured the cosine-similarity between representations for three types of image pairs (Fig. 4A): (1) Same-parts/different-category: anagram pair composed of images sharing identical puzzle pieces but representing different global categories (e.g., wolf vs. elephant); (2) Different-parts/different-category: with containing different puzzle pieces and an equivalent categorical difference as a matched anagram pair (e.g., wolf vs. elephant); and (3) Different-parts/same-category pairs composed of different puzzle pieces but the same category (e.g., both wolves). We evaluated 60 image pairs of each type (180 total). If cosine-similarity depends on shared puzzle pieces, we would expect a higher correlation for the anagram pair (same-parts, different-category) than for either of the other pairs (which all have different parts). If cosine-similarity depends only on similarity in global-configuration (category), then it should be higher for the Different-Parts/Same-Category pairs and equally low for the the other pairs (which all have different category). Based on the ablation study, we expect high-CSS models to show configural effects emerging by middle-to-late model layers.

We quantified representational similarity using cosine similarity between image embeddings at each intermediate layer of a model. Fig. 4B shows layer-by-layer results for one selected high-CSS model (EVA-CLIP G/14 ( CSS=77.78%), and one-selected low-CSS model (Resnet50, CSS=16.67%). Focusing first on the high-CSS model (top), several patterns emerge that are consistent with the idea that configural shape representations emerge in intermediate layers and dominate the final output of the model. First, in early model layers, local-similarity dominates: image pairs with shared parts (green) are more similar to each other than image pairs with different parts (orange and blue). Second, just beyond the midpoint, the effect of category similarity emerges: images with the different-parts/same-category (orange) begin to show greater similarity than the different-parts/different-category pairs (blue), and by the later layers the same-category pairs actually show *greater* similarity than the anagram pair (green). Indeed, by the final layers, the green/blue lines have collapsed together, indicating that having the same puzzle pieces is irrelevant by that point, and only the configural/category-level similarity matters. The results for the low-CSS Resnet50 are markedly different, and suggest that the ResNet50 model never shows a transition to more configuration-based processing. As shown in Fig 4B (bottom), the ResNet50 model shows greater part-based similarity (green) across all layers, including the final layers, and there is only a slight increase in similarity for

the different-part/same-category pairs (orange) relative to the different-part/different-category pairs (blue) at the final layer.

We formalized these qualitative observations by computing two metrics (Fig. 4C) – Puzzle component influence and Category influence – over the last and penultimate layer of all models. Puzzle component influence is the difference in cosine similarity between pairs with identical puzzle pieces (anagram pairs) and those with different puzzle pieces and the matched category differences (different-parts/different-category). Category Influence is the average similarity advantage for same-category pairs over different-category pairs, irrespective of local puzzle pieces. We then measure whether the configural shape scores can predict these metrics across all the models. Fig. 5C shows this relationship for the last layer. Higher Configural Shape Score corresponded to lower Puzzle Component Influence (negative correlations: $r = -0.70$ at the last layer, $r = -0.57$ at the penultimate layer), suggesting that higher-CSS models focus less on details of the local part appearance. Conversely, Configural Shape Score correlated positively with Category Influence ($r = 0.80$ at the last layer, $r = 0.83$ at the penultimate layer), indicating that high-CSS models encode representations that are shared between images within a category, while discriminating between categories. This trend is observed even when considering representations from DINOv2 backbones, which are fully self-supervised and have no pressure to form abstract category representations ($r = 0.94$ between CSS and Category Influence and $r = -0.86$ between CSS and Puzzle Component Influence).

Taken together with the results of the ablation study, these results suggest that long-range contextual interactions enable high-CSS models to transition from representations that are initially dominated by local parts, to representations that are dominated by a holistic view that depends on the configuration of parts — i.e., encodes the image as more than the sum of its parts, specifically in terms of relationships between those parts.

## 9   BagNets Provide Evidence Against a "Border Hacking" Solution

Can a local strategy be used to recognize both pairs of images in a visual anagram? When rearranging and rotating the puzzle pieces, what if features that emerge at the intersection between abutting pieces are sufficient to identify the global category of each image in a pair, yielding successful classification through non-configural "border-hacking"? BagNets [25] provide some evidence against the viability of a local solution for anagram recognition. These models have a ResNet-style architecture, except that the receptive field of the intermediate units is highly restricted (at max 9, 17, or 33 pixels throughout), making them very sensitive to local features and incapable of any kind of long-range spatial/contextual interaction. While they are competent in ImageNet recognition (with top1 accuracy: Bagnet9 at 41.38%, Bagnet17 at 55.08%, Bagnet33 at 61.28%), they all have very low CSS (Bagnet9 at 2.78%, Bagnet17 at 1.38%, Bagnet33 at 5.5%). Overall, the near-chance CSS of Bagnets underscores that fine-scale junction statistics alone are insufficient for anagram disambiguation, strengthening the interpretation of CSS as a global-configuration probe.

## 10   From Configural Shape Score to Broad Shape-Dependent Performance

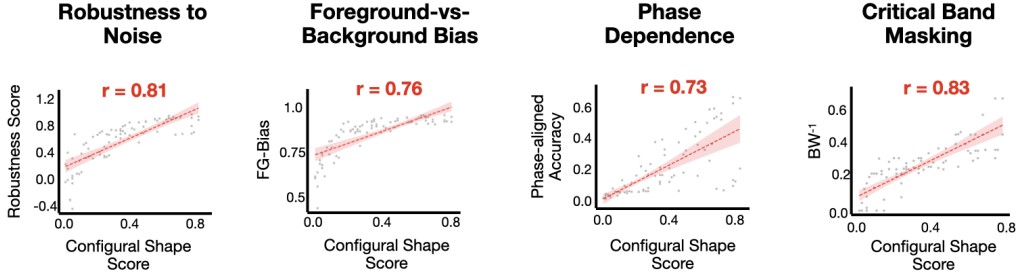

Figure 5: **Configural Shape Score (CSS) predicts model performance across a range of benchmarks.** CSS is positively correlated with foreground-vs-background bias, robustness to noise, phase dependence and critical band masking bandwidth

To what degree does having a high CSS predict other representational benefits and qualities? As shown in Fig. 5, we found that Configural Shape Score was significantly correlated with scores from several evals, including: 1) Robustness to Noise (r=0.81), testing each model's performance across varying severity levels for five distinct noise types, as described in [11]. 2) Foreground-vs-Background

Bias (r=0.76), tested using ImageNet-9 dataset from [68], quantifying the extent to which a model relies primarily on the foreground object rather than background information for classification. 3) Phase Dependence (r=0.73), assessed by swapping phase information in Fourier space between images and then measuring top-1 accuracy, quantifying the model's reliance on phase information [69]. 4) Critical band masking strategy outlined by [14] (r=0.83), used to determine the bandwidth of spatial frequencies essential for accurate object recognition. In contrast, the shape-vs-texture bias score across these models showed weaker relationships to these evals (r=0.62 with Robustness to Noise; r=0.32 with Foreground-vs-Background Bias; r=0.52 with Phase Dependence and r=0.55 with Critical Band Masking). Statistical comparison using Williams's test confirmed that CSS was a significantly better predictor of these metrics than shape-vs-texture bias (all p<0.001; see A.7 for test statistics and details). These results suggest that models with better configural shape scores also have other favorable and human-like perceptual qualities. See A.6 for more information about these evaluations, and A.8 and A.9 for feature attributions to qualitatively compare low- and high-CSS models on challenging stimuli from each benchmark.

## 11 Limitations and Discussion

Although the Object Anagram Dataset and the accompanying Configural Shape Score (CSS) provide a quantitative measure of holistic processing, several caveats warrant mention. First, the stimuli we generate are constrained by the priors of the diffusion model, and may explore only a subspace of configural encoding relationships. Furthermore, we use a uniform "black paint texture" to ensure local cues are perfectly matched, though this raises the possibility of out-of-distribution effects. However, our analyses show that even models with poor CSS can classify single anagram images successfully, suggesting the primary failure is in configural processing, not in handling the texture itself. Second, this work targets whole-object configurations and therefore does not directly probe part-based compositionality, an orthogonal facet of shape reasoning that future work should address. Third, despite containing thousands of composites, the dataset is modest in scale compared with modern billion-image corpora, suggesting that larger or more ecologically varied stimuli could reveal subtler effects. To that end, we created a 20x larger set of 1440 anagram pairs to test the robustness of our main findings. Re-evaluating all 86 models confirmed that the CSS metric is highly stable: the relative model rankings were preserved, and the scores from the original and expanded datasets showed a correlation of r=0.99 (see Appendix A.10).

Within these bounds our contributions are threefold. First, we formalized configural shape sensitivity as an interpretable metric distinct from texture reliance. Second, by charting CSS across 86 pretrained networks, we showed that holistic competence is neither fully explained by ImageNet accuracy nor canonical shape-vs-texture bias. Third, attention ablation and representational similarity analyses revealed that CSS relies on intermediate-stage, long-range interactions enabling recognition based on the configuration of parts and contextual relations. Finally, we demonstrated that models with higher CSS also perform better across other shape-dependent evaluations. These findings highlight configural shape processing as a critical yet underexplored dimension of visual intelligence and invite future work in advancing vision models toward human-like holistic representations.

Our results suggest that high configural competence stems primarily from training objectives that enforce local-to-global consistency. For instance, DINOv2's teacher-student objective, which trains a student network on local image crops to match the output distribution of a teacher network seeing global crops, directly rewards the integration of long-range information. This view-consistency objective leads to a more powerful signal than standard data augmentations alone or purely reconstructive approaches like MAE, whose local pixel-prediction task is insufficient on its own to produce the mid-layer "configural flip" characteristic of high-CSS models. Language supervision (BEiT-v2, EVA-CLIP, CLIP, SigLIP) would also enforce alignment between standard local-image crops and a global text caption, and this form of local-global consistency objective could also place an emphasis on multiple distant concepts in the image. Together, these findings point to a clear roadmap for building more shape-aware systems, which should prioritize (1) global-local consistency objectives, paired with (2) architectures capable of long-range integration, and (3) evaluated with explicit shape metrics like CSS.

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

# A    Appendix

## A.1    Shape-vs-Texture Bias: A Useful but Incomplete Metric for Assessing Shape Representations

A widely used benchmark for assessing the degree to which models rely on shape is shape bias, introduced by Geirhos et al. (2019). In this paradigm, each stimulus is a hybrid image that contains the shape defined by one category (e.g., the shape of a cat) with a conflicting texture from a different category (e.g., elephant skin). Humans show strong shape preference in this task, performing near ceiling ( 95%).In contrast, standard deep net models such as ResNet-50 typically exhibit a strong texture bias, favoring the incongruent texture on  70–80% of trials. While this paradigm reflects the model's relative preference between two competing cues—shape or texture—it is ambiguous if a high score is attained by suppressing textural information or enhancing shape representations. For all shape-vs-texture bias we follow the updated method used in [18] that adjusts for baseline shape accuracy, providing a more principled measure of shape quality using the following equation:

$$\text{Shape-vs-Texture Bias} \atop \text{(accuracy-corrected)} = \sqrt{\frac{\text{\#Correct Shape Decisions}}{\text{\#Correct (Shape + Texture)}}} \times \sqrt{\frac{\text{\#Correct Shape Decisions}}{\text{Total Trials}}}$$

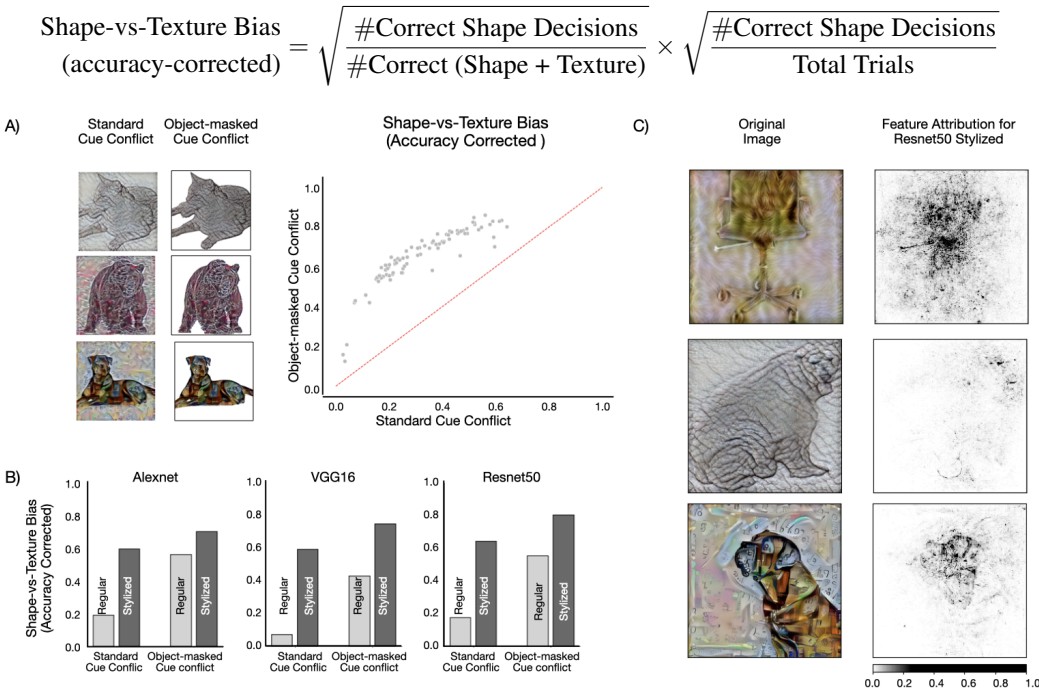

Figure 6: (A) (Left) Cue-conflict stimuli illustrating hybrid images composed of shape from one category and texture from another. Object-masked cue-conflict stimuli has texture removed from the background (Right) Shape-vs-Texture bias (accuracy corrected) across models using standard and object-masked cue conflict stimuli. (B) Shape-vs-Texture bias (accuracy corrected) for original and stylized models. (C) Feature attribution on ResNet50-Stylized reveals that model decisions are still driven by local texture-rich patches, not the full object extent, suggesting that stylization does not completely enhance shape processing.

In Fig. 6A we measure shape-vs-texture bias for a variant of the cue conflict dataset developed [70], in which the conflicting textures are masked out in the background, preserving only the object silhouette. If shape-vs-texture bias truly reflects shape representations, this manipulation should not significantly alter bias scores. However, across a broad range of well-trained deep networks (n = 86), we observed consistent and substantial increases in bias scores under the object-masked condition. To contextualize these findings, we compared the improvements achieved by background masking with those achieved through stylization-based training. As shown in Fig. 6B, stylized models evaluated on the standard cue conflict task showed as much shape-vs-texture bias gains as it would have shown when vanilla (non-stylized) architectures were tested on object-masked cue-conflict stimuli, suggesting that the shape bias measure is confounded not only by texture within

the object but also by surrounding local image statistics that lie outside the object's boundary. In Fig. 6C, we applied attribution-based analyses on ResNet50-Stylized to assess which parts of the image were most influential for the model's decision. The results show that model activations often remained highly localized—focusing on small, texture-rich fragments—rather than spanning the full extent of the object, consistent with findings from [19]. In other words, these results suggest that merely suppressing texture either during training (i.e. via stylization) or removing texture footprint in images via silhouette masking, all while keeping the shape information intact, can drive a model's shape-vs-texture bias scores up. Together, these findings suggest that while shape bias remains a valuable comparative diagnostic for assessing model preference between competing shape and texture cues, it should not be interpreted as the only single evidence of good quality shape representation.

## A.2 Compute Details

All models were analyzed on an internal computer cluster with 24 cores, 384GB of system RAM, and a NVIDIA H100 GPU with upto 80GB memory. The object Anagram Dataset was generated using a single NVIDIA A100 GPU with 40 GB memory.

## A.3 Extracting 1000-way logits and mapping to 9 categories from the Object Anagram Dataset

For self-supervised models like BeITs, MAEs, CLIP, and EVA-CLIP models we used the finetuned linear classifier head provided via the timm library and for DinoV2 we used the full-4 classifier head provided with the model backbone. For SigLIP models we analyzed zero-shot predictions extracted by probing the outputs of the vision encoder with embeddings from text encoder using the category prompts and the given image as inputs.

To map the 1000-way ImageNet logits to our nine target categories in the Object Anagram Dataset, we used the category-to-ImageNet class mapping used in [9] (see below). For each target category, we collected logits corresponding to each category's ImageNet class indices and then took the maximum value from those indices. Once a logit value was computed for each target category, we applied a softmax to get a 9-way probability vector. The predicted label was set to the category with the highest probability.

| Category | ImageNet Class Indices |
|---|---|
| bear | [294, 295, 296, 297] |
| bunny | [330, 331, 332] |
| cat | [281, 282, 283, 284, 285] |
| elephant | [101, 385, 386] |
| frog | [30, 31, 32] |
| lizard | [38, 39, 40, 41, 42, 43, 44, 45, 46, 47, 48] |
| tiger | [286, 287, 288, 289, 290, 291, 292, 293] |
| turtle | [33, 34, 35, 36, 37] |
| wolf | [269, 270, 271, 272, 273, 274, 275] |

Table 2: Mapping between 9 target categories on Object Anagram Dataset and their corresponding ImageNet class indices.

## A.4 Human Configural Shape score Estimate

We measured human Configural Shape Score (CSS) using a behavioral experiment implemented in jsPsych. Participants first completed informed consent and viewed instructions explaining the task and rationale. Each trial began with a fixation display followed by an image from the dataset presented centrally for 750 milliseconds. Immediately afterward, a noise mask consisting of randomly generated grayscale pixels appeared for 500 milliseconds to disrupt visual persistence. Following the mask, participants selected the object's category from nine visually presented icons (bear, bunny, cat, elephant, frog, lizard, tiger, turtle, or wolf). Participants completed all 144 images (72 anagram pairs), presented in randomized order, with their category selections and response times recorded. The resulting human CSS served as an approximate baseline for evaluating the configural shape sensitivity of the computational vision models. This study was approved by the IRB of the corresponding author's home institution.

## A.5 Cumulative Ablation Analysis on DINOv2 Models

To more precisely characterize the role of intermediate layers for configural processing, we performed a cumulative ablation study on the DINOv2 model family (S/14, B/14, L/14, and G/14) and the ViT-B/16 baseline. We hypothesized that the single-block ablation presented in Section 6 might be too subtle to reveal the full processing dynamics in deeper models. To address this, we designed an ablation experiment with two conditions: (1) ablating all blocks up to block 'n' to isolate the contribution of the network up to that point, and (2) ablating all blocks after block 'n' to test for recovery by later layers. The ablation follows the same strategy of masking the attention in the targeted blocks using a mask window of 2 nearby patches for all the patches, hence blocking long-range interactions.

Fig. 7 shows the effect of these ablations on the final class token representation, measured by its cosine similarity to the token from the original, unablated model. In the "Ablate all before" condition (blue line), the similarity starts high and drops steeply as intermediate blocks are removed, while the "Ablate all after" condition (orange line) shows the opposite trend, creating the characteristic crossover point in the intermediate layers of each model. The results demonstrate that across all DINO models tested, the drop in CSS is highest and most irrecoverable in the intermediate layers. This confirms that even for the large and giant DINO variants, the mid-depth layers are the critical locus for integrating local features into a global, configural shape representation.

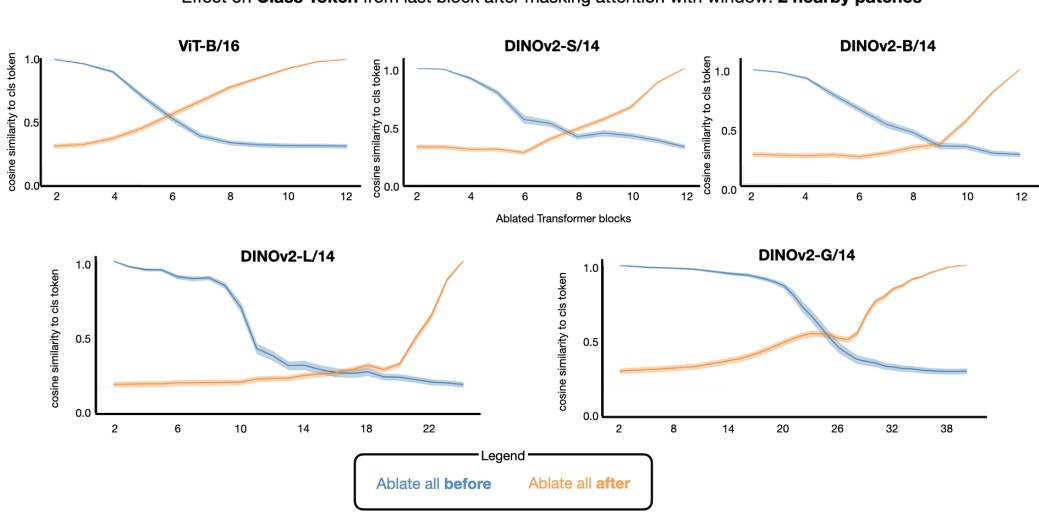

Figure 7: **Effect of cumulative ablation on class token representation.** The y-axis shows the cosine similarity between the final class token of the original model and the ablated model. The blue line (*Ablate all before*) shows the condition when ablating long-range interactions that is 'accumulated' in all blocks before block n. The orange line (*Ablate all after*) shows the condition when ablating long-range interactions that could be 'recovered' in all blocks after block n.

## A.6 Other Shape-dependent Evals

| Evaluation | Stimuli | # Images | # Categories | Ref. |
|---|---|---|---|---|
| Robustness to Noise | Imagenette2 | 98,125 | 10 | Hendrycks & Dietterich, 2019 (and fastai) [11] |
| Foreground Bias | Imagenet9 | 4050 | 9 | Xiao et al., 2020 [68] |
| Shape-vs-Texture Bias | Cue Conflict | 1200 | 16 | Geirhos et al., 2018 [8, 18] |
| Critical Band Masking | Imagenet | 1050 | 16 | Subramanian et al., 2023 [14] |
| Phase-Dependence | Imagenet | 50k | 1000 | Garity et al., 2024 [69] |

Table 3: Overview of evaluation metrics, stimuli, and dataset statistics.

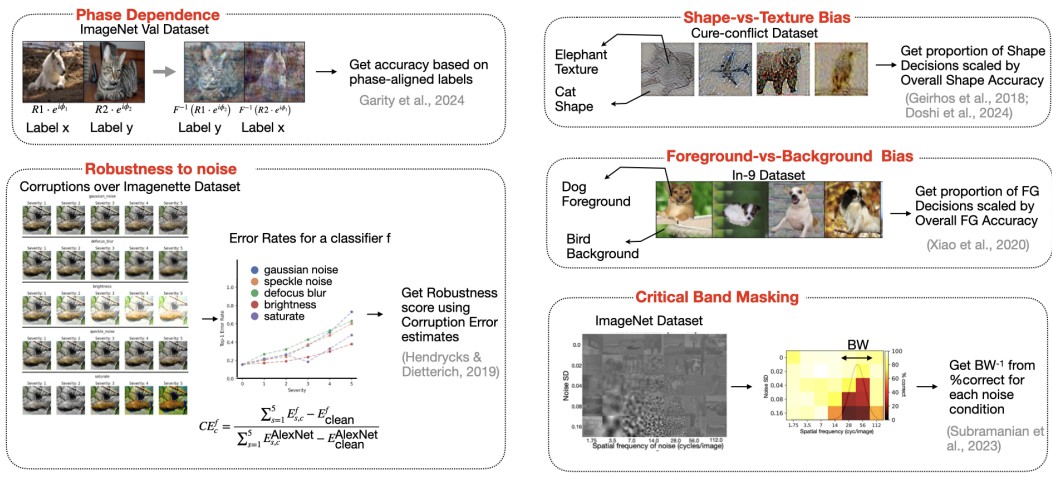

Figure 8: **Schematic of other Shape-dependent Evals.**

## A.7 Statistical Comparison of Predictive Strength: CSS vs. Shape-vs-Texture Bias

To evaluate whether Configural Shape Score (CSS) better predicts other shape-dependent evals than the traditional Shape-vs-Texture Bias, we used Williams's test for dependent correlations with one variable in common (i.e., each eval score). The test compares two correlation coefficients (corr(CSS, Eval) and corr(Shape-vs-Texture Bias, Eval)) that share a common outcome variable, accounting for the correlation between the two predictors. We tested this for each of the four evals: Robustness to Noise, Foreground-vs-Background Bias, Phase Dependence, and Critical Band Masking Bandwidth.We used a one-tailed Williams test with n = 86 models, reflecting the directional hypothesis that CSS should better predict eval performance than Shape-vs-Texture Bias. All tests were statistically significant at p < 0.01, indicating that Configural Shape Score is a significantly stronger predictor of these eval benchmarks than Shape-vs-Texture Bias. Below are the results:

| Eval Benchmark | r(CSS, Eval) | r(Shape-vs-Texture Bias, Eval) | t-value | p-value |
|---|---|---|---|---|
| Robustness to Noise | 0.81 | 0.62 | 3.4116 | 0.0005 |
| Foreground-vs-Background Bias | 0.76 | 0.32 | 7.618 | <0.0001 |
| Phase Dependence | 0.73 | 0.52 | 3.39 | 0.00053 |
| Critical Band Masking | 0.83 | 0.55 | 5.47 | <0.0001 |

Table 4: Comparison of Configural Shape Score (CSS) and Shape-vs-Texture Bias.

## A.8 Feature Attribution for Challenging stimuli

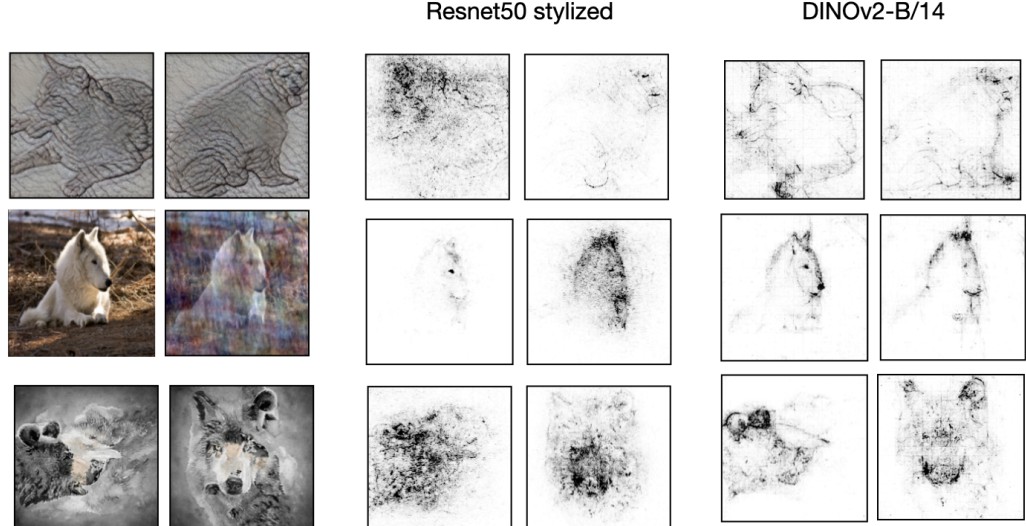

Figure 9: **Feature Attribution Maps for challenging stimuli.** Maps generated using Integrated Gradients for cue-conflict stimuli, phase swapped stimuli, and visual anagrams.

## A.9 Feature Attributions of Anagrams in High-CSS model (DINOv2-B/14) are not Anagrams themselves

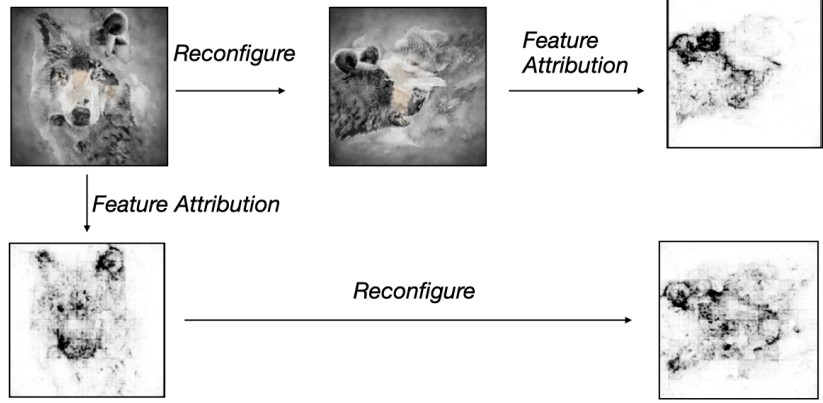

Figure 10: **Applying permutation transformation on feature attribution maps of visual anagrams.**

## A.10  Validation on Expanded Anagram Dataset

A potential limitation of our initial study was the dataset size (72 anagram pairs). To address this, we expanded the dataset by 20x to include 1440 visual-anagram pairs and re-evaluated all 86 vision models. The strong correlation (r=0.99) between the scores on the original and expanded datasets shown in Fig. 11 demonstrate that the Configural Shape Score is a highly stable measurement and the conclusions drawn in the main paper are robust.

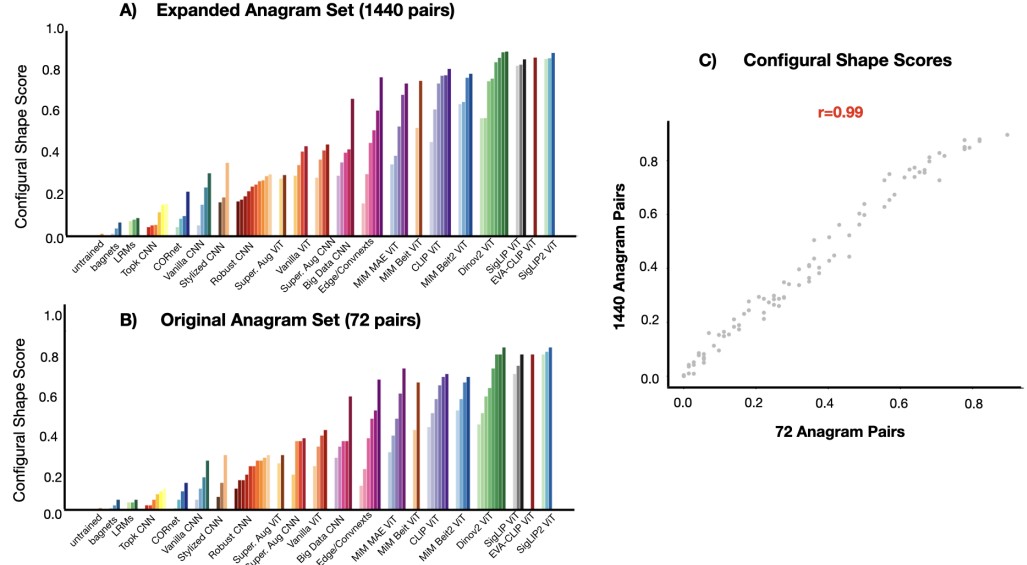

Figure 11: **Validation of Configural Shape Scores (CSS) on an expanded dataset.** (A) CSS scores for all models on the expanded 1440-pair anagram set. (B) The original CSS scores on the 72-pair set. (C) A scatter plot between the scores from the two datasets.

## A.11  Evaluated Models

| Architecture | Notes | Type |
|---|---|---|
| AlexNet | Untrained baseline | Standard Convolutional Networks (TorchVision) |
| VGG-16 | Untrained baseline | Standard Convolutional Networks (TorchVision) |
| ResNet-50 | Untrained baseline | Standard Convolutional Networks (TorchVision) |
| ResNet-101 | Untrained baseline | Standard Convolutional Networks (TorchVision) |
| AlexNet | Standard AlexNet | Standard Convolutional Networks (TorchVision) |
| VGG-16 | Standard VGG-16 | Standard Convolutional Networks (TorchVision) |
| ResNet-50 | Supervised baseline | Standard Convolutional Networks (TorchVision) |
| ResNet-101 | Supervised baseline | Standard Convolutional Networks (TorchVision) |
| ViT-B/16 | Base | Standard supervised Vision Transformers (TorchVision) |
| ViT-L/16 | Large | Standard supervised Vision Transformers (TorchVision) |
| ViT-B/32 | Base | Standard supervised Vision Transformers (TorchVision) |
| ViT-L/32 | Large | Standard supervised Vision Transformers (TorchVision) |
| AlexNet | Shape-biased AlexNet | Stylized Models |
| ResNet-50 | Shape-biased ResNet-50 | Stylized Models |
| VGG-16 | Shape-biased VGG-16 | Stylized Models |
| ResNet-50 | L2 $\varepsilon = 0$ | Adversarially Robust Models |
| ResNet-50 | L2 $\varepsilon = 0.01$ | Adversarially Robust Models |
| ResNet-50 | L2 $\varepsilon = 0.03$ | Adversarially Robust Models |
| ResNet-50 | L2 $\varepsilon = 0.05$ | Adversarially Robust Models |
| ResNet-50 | L2 $\varepsilon = 0.1$ | Adversarially Robust Models |
| ResNet-50 | L2 $\varepsilon = 0.25$ | Adversarially Robust Models |
| ResNet-50 | L2 $\varepsilon = 0.5$ | Adversarially Robust Models |
| ResNet-50 | L2 $\varepsilon = 1.0$ | Adversarially Robust Models |
| ResNet-50 | L2 $\varepsilon = 3.0$ | Adversarially Robust Models |
| ResNet-50 | L2 $\varepsilon = 5.0$ | Adversarially Robust Models |
| Alexnet (top-k=80%) | Sparse activation variant | Top-k Sparse |

| Architecture | Notes | Type |
|---|---|---|
| Alexnet (top-k=60%) | Sparse activation variant | Top-k Sparse |
| Alexnet (top-k=40%) | Sparse activation variant | Top-k Sparse |
| VGG-16 (top-k=80%) | Sparse activation variant | Top-k Sparse |
| VGG-16 (top-k=60%) | Sparse activation variant | Top-k Sparse |
| VGG-16 (top-k=40%) | Sparse activation variant | Top-k Sparse |
| Resnet50-BagNet-9 | Local Receptive field limited to 9 pixels | BagNets |
| Resnet50-BagNet-17 | Local Receptive field limited to 17 pixels | BagNets |
| Resnet50-BagNet-33 | Local Receptive field limited to 33 pixels | BagNets |
| ResNet-50 | Instagram-1B → ImageNet | Data-scale ResNets (SWSL / SSL) |
| ResNet-50 | YFCC100M → ImageNet | Data-scale ResNets (SWSL / SSL) |
| Resnet-50v2 | BiT | Data-scale ResNets (BiT) |
| Resnet-50v2 | BiT-distilled (ResNet-101x1) | Data-scale ResNets (BiT) |
| Resnet-101v2 | BiT | Data-scale ResNets (BiT) |
| ResNet-50 | Supervised with strong augmentation | Strong Augmentation Baselines (Timm) |
| ResNet-101 | Supervised with strong augmentation | Strong Augmentation Baselines (Timm) |
| ViT-S/16 | Supervised with strong augmentation (Small) | Strong Augmentation Baselines (Timm) |
| ViT-B/16 | Supervised with strong augmentation (Base) | Strong Augmentation Baselines (Timm) |
| ConvNeXt-Base | Modern CNN | ConvNeXt |
| ConvNeXtV2-Base | FCMAE→ImageNet | ConvNeXt |
| ConvNeXtV2-Huge | FCMAE→ImageNet | ConvNeXt |
| EfficientFormer-L1 | Efficient transformer, SnapDist | EfficientFormer |
| EfficientNet-B0 | JFT Pretraining | EfficientNet |
| Edge-AlexNet | Trained on edge-filtered images | Edge-trained Alexnet |
| CORnet-Z | CORnet Family | Bio-Inspired |
| CORnet-R | CORnet Family | Bio-Inspired |
| CORnet-RT | CORnet Family | Bio-Inspired |
| CORnet-S | SCORnet Family | Bio-Inspired |
| Alexnet-LRM-Pass1 | Long-Range Modulatory CNNs 1 | Bio-Inspired |
| Alexnet-LRM-Pass2 | Long-Range Modulatory CNNs 1 | Bio-Inspired |
| Alexnet-LRM-Pass3 | Long-Range Modulatory CNNs 1 | Bio-Inspired |
| BEiT-B/16 | ImageNet-22K → ImageNet | Self-supervised ViT BEiT |
| BEiT-L/16 | ImageNet-22K → ImageNet | Self-supervised ViT BEiT |
| BEiTv2-B/16 | ImageNet | Self-supervised ViT BEiTv2 |
| BEiTv2-L/16 | ImageNet | Self-supervised ViT BEiTv2 |
| BEiTv2-B/16 | ImageNet-22K → ImageNet | Self-supervised ViT BEiTv2 |
| BEiTv2-L/16 | ImageNet-22K → ImageNet | Self-supervised ViT BEiTv2 |
| Hiera-MAE-T | Hierarchical MAE (tiny) | Self-supervised ViT MAE |
| Hiera-MAE-S | Hierarchical MAE (small) | Self-supervised ViT MAE |
| Hiera-MAE-B | Hierarchical MAE (base) | Self-supervised ViT MAE |
| Hiera-MAE-L | Hierarchical MAE (large) | Self-supervised ViT MAE |
| Hiera-MAE-H | Hierarchical MAE (huge) | Self-supervised ViT MAE |
| DINOv2-ViT-S/14 | DINO2 (small) | Self-supervised ViT DINOv2 |
| DINOv2-ViT-B/14 | DINO2 (base) | Self-supervised ViT DINOv2 |
| DINOv2-ViT-L/14 | DINO2 (large) | Self-supervised ViT DINOv2 |
| DINOv2-ViT-G/14 | DINO2 (giant) | Self-supervised ViT DINOv2 |
| DINOv2-ViT-S/14 | DINO2 (small) + 4 Registers | Self-supervised ViT DINOv2 |
| DINOv2-ViT-B/14 | DINO2 (base) + 4 Registers | Self-supervised ViT DINOv2 |
| DINOv2-ViT-L/14 | DINO2 (large) + 4 registers | Self-supervised ViT DINOv2 |
| DINOv2-ViT-G/14 | DINO2 (giant) + 4 Registers | Self-supervised ViT DINOv2 |
| CLIP ViT-B/16 | OpenAI CLIP (base) | Language-aligned ViT CLIP |
| CLIP ViT-B/32 | OpenAI CLIP (base) | Language-aligned ViT CLIP |
| CLIP ViT-L/14 | OpenAI CLIP (large) | Language-aligned ViT CLIP |
| CLIP ViT-L/14 | OpenAI CLIP (large) → ImageNet-12K | Language-aligned ViT CLIP |
| CLIP ViT-H/14 | LAION-2B CLIP (huge) | Language-aligned ViT CLIP |
| CLIP ViT-H/14 | LAION-2B CLIP (huge) → ImageNet-12K | Language-aligned ViT CLIP |
| SigLIP ViT-B/16 | SigLIP (base), Image size 224, zeroshot | Language-aligned ViT SigLIP |
| SigLIP ViT-B/16 | SigLIP (base), Image size 256, zeroshot | Language-aligned ViT SigLIP |
| SigLIP ViT-L/16 | SigLIP (large), Image size 256, zeroshot | Language-aligned ViT SigLIP |
| SigLIP2 ViT-B/16 | SigLIP2 (base), Image size 224, zeroshot | Language-aligned ViT SigLIP2 |
| SigLIP2 ViT-B/16 | SigLIP2 (base), Image size 256, zeroshot | Language-aligned ViT SigLIP2 |
| SigLIP2 ViT-L/16 | SigLIP2 (large), Image size 256, zeroshot | Language-aligned ViT SigLIP2 |
| EVA-CLIP-G/14-g | EVA02 CLIP (giant) | Language-aligned ViT EVA-CLIP |

