# OpenReview forum: "Visual Anagrams Reveal Hidden Differences in Holistic Shape Processing Across Vision Models"
_NeurIPS.cc/2025/Conference — NeurIPS 2025 poster_

### Official Review · Reviewer_iP6z · 2025-07-02

**Clarity:** 2
**Significance:** 3
**Originality:** 2
**Rating:** 5
**Confidence:** 3

**Summary:**

The authors introduce the Configural Shape Score (CSS), an absolute metric for holistic shape processing on “visual anagram” pairs. They show that transformer-based vision models reach near-human CSS via learning long-range interaction and that CSS predicts model robustness better than classic bias measures.

**Questions:**

- How many human participants were recruited for the study? This should be clearly stated.
 - Have the authors plotted “Top1 ImageNet Accuracy” vs. “Shape-vs-Texture Bias”? I believe a plot like this could be interesting if it is shown next to the “Top1 ImageNet Accuracy” vs. “CSS” plot in Figure 2. Would the R^2 be lower for Shape-vs-Texture Bias?

**Ethical Concerns:**

["NO or VERY MINOR ethics concerns only"]

**Final Justification:**

The authors resolved my concerns regarding clarity and small sample size, and I cannot think of any more flaws or shortcomings in the paper. This work is novel and well-written.

**Limitations:**

yes

**Quality:**

3

**Strengths And Weaknesses:**

**Quality (Good):**

This paper is technically sound, and the main results were well supported by additional experiments (ablation and disentanglement analyses). The experiments are extensive, although I do agree with the authors that they could have (perhaps marginally) benefited from creating a larger dataset.


**Clarity (Fair):**

The authors did well in establishing the gap in the literature and how their work bridges this gap. Overall, the paper is easy to read. However, the abstract can be substantially improved by making it more succinct. Currently, the abstract (especially its latter part) is only understandable if you have read the entire manuscript, which defies the purpose of having an abstract.


**Significance (Good):**

I believe this dataset and CSS can/should be utilized by many vision researchers in the future. The utility of the authors’ framework is very well supported by the experimental results.


**Originality (Fair):**

Applying the vision-anagram technique for understanding shape vs texture processing is novel and useful.

---

> ### Author Rebuttal · Authors · 2025-07-30
>
> Thank you for your positive feedback on the quality and significance of our work. We truly appreciate your statement that “this dataset and CSS can/should be utilized by many vision researchers in the future,” and we wholeheartedly agree with this perspective.
>
> ---
>
> >  **“The abstract can be substantially improved by making it more succinct.”**
>
> Yes, we agree. We'll revise the abstract to make it more succinct and group the key findings more clearly. We will edit the abstract to be shorter and more clear in the camera ready version.
>
> ---
>
> >  **“The experiments are extensive, although I do agree with the authors that they could have (perhaps marginally) benefited from creating a larger dataset.”**
>
> You're absolutely right, and we agree that a larger dataset strengthens the findings. We've already expanded the corpus to 1440 visual anagram pairs, a 20x increase, and re-evaluated all models. The results are highly consistent: model scores remain stable, and the relative CSS rankings of the models are preserved, the correlation between the CSS scores on the original and expanded datasets is r = 0.99. These updated results and the full dataset will be included in the camera-ready version.
>
> ---
>
> > **“How many human participants were recruited for the study? This should be clearly stated.”**
>
> Only two participants were tested, as the goal of this study was only to confirm that the images in anagram pairs are easily recognizable. These images are not meant to to challenge human recognition, and the small sample already verify this.
> Moreover, the results are not intended for making direct comparisons between model and human behavior, which would require a larger-N (Alvarez & Konkle, 2024). We will clarify the role of these behavioral data in the manuscript.
>
> ---
>
> >  **Have the authors plotted “Top1 ImageNet Accuracy” vs. “Shape-vs-Texture Bias”? I believe a plot like this could be interesting if it is shown next to the “Top1 ImageNet Accuracy” vs. “CSS” plot in Figure 2. Would the R^2 be lower for Shape-vs-Texture Bias?**
>
> We found that adding an extra subplot to the Figure 2 disrupted the clarity a bit, but we instead will put a new figure in the Appendix, which also will show all pairwise relationships among CSS, Shape-vs-Texture Bias, Top-1 Accuracy, and related metrics.  Thank you for your suggestion.

---

> > ### Comment · Reviewer_iP6z · 2025-08-04
> >
> > Thank you for addressing my concerns. I have raised the score to 5.

---

### Official Review · Reviewer_3trt · 2025-07-02

**Clarity:** 3
**Significance:** 2
**Originality:** 2
**Rating:** 4
**Confidence:** 5

**Summary:**

This paper introduces the Configural Shape Score (CSS), an evaluation metric that assesses a vision model's sensitivity to global shape configuration by using visual anagram image pairs—images constructed from the same set of patches but rearranged to represent different object categories. The authors show that CSS correlates with multiple shape-related benchmarks and is highest in self-supervised and language-aligned vision transformers (e.g., DINOv2, EVA-CLIP, SigLIP). Mechanistic analyses (attention ablations and representational similarity) reveal that high-CSS models rely on long-range attention and develop mid-layer configural representations.

**Questions:**

- Given the strong correlations between CSS and prior shape-related metrics (e.g., robustness, shape bias, foreground-background bias), can you elaborate on what new diagnostic power CSS offers? What specific types of failure modes or model differences would CSS detect that previous methods would miss?
- Can you evaluate CNN-based self-supervised models such as MoCo, SimCLR, BYOL, or SwAV to determine whether high CSS requires attention, or if it can arise from training objectives and augmentations alone?

- Did you try training or fine-tuning DINOv2 (or a similar ViT) without spatial augmentations (e.g., removing random cropping or multi-view augmentations) to assess whether positional robustness is necessary for high CSS?

- Given that your visual anagram patches are larger than ViT patch size (64×64 vs. 16×16), do you think CSS performance reflects models' ability to integrate subpatch relationships, or is it primarily a function of spatial layout processing?

- Can you clarify whether the representational similarity analyses provide insight beyond the expected hierarchical abstraction of deep networks (i.e., early layers being input-like, later layers being task-driven)?

- Have you tested models with different types of position encodings (e.g., relative vs. absolute, or no encoding) to better understand whether CSS relies on absolute spatial anchoring or on relational structure?

**Ethical Concerns:**

["NO or VERY MINOR ethics concerns only"]

**Final Justification:**

I have read the authors' responses to reviews, and I appreciate the clarifications and the few experiments added to the supplementary. I am not however convinced to increase the score, because CSS while visually pleasing and intriguing doesn't provide any further insight into mechanisms of visual processing compared to the existing methods which it was compared to in the paper. However, this line of work is promising, and I am optimistic that next versions of the work could provide such mechanistic insights.

**Limitations:**

Yes

**Quality:**

3

**Strengths And Weaknesses:**

### Strengths

- **Clever, well-controlled benchmark**: The visual anagram paradigm is a creative way to leverage diffusion models to synthesize image sets and isolate configural shape sensitivity, controlling for texture completely.


### Weaknesses

- **Lack of novelty in insight**: CSS is strongly correlated with existing shape-sensitive metrics (e.g., robustness, foreground bias, shape-vs-texture bias), and the core mechanistic findings (need for long-range integration, mid-layer configural transitions) are consistent with prior literature and likely generalize to other shape-related evaluations.
- **Better use of mechanistic analysis: pixel vs latent self-supervised**: While MAEs are included in the CSS benchmark and perform moderately, the paper does not explore *why* they underperform compared to models like DINOv2. This is a missed opportunity. MAEs are self-supervised but trained with a pixel-level reconstruction objective, which likely enforces precise spatial encoding and discourages position-invariant configural representations. A mechanistic analysis—such as attention ablation or representational similarity profiling—would help clarify whether MAEs fail to develop global shape sensitivity due to architectural limitations or due to the nature of the objective itself. Including such an analysis would strengthen the argument that not all self-supervision leads to configural competence.
- **Augmentation not disentangled**: High CSS may be primarily driven by the use of strong spatial augmentations during training (e.g., cropping in DINOv2), but this is not tested directly. A simple ablation (e.g., DINOv2 w/o crop) would help isolate the contribution of augmentation vs. architecture.
- **Unclear advantage of the mechanistic analyses** The mechanistic analyses (e.g., layer-wise representational similarity) show that high-CSS models transition from part-based to configural representations across layers. However, this finding may be somewhat tautological: given that CSS is a classification output metric, it's expected that later layers — which drive the prediction — will reflect configural coding, while early layers reflect lower-level inputs. This hierarchical progression is a general property of deep networks and does not, in itself, constitute a novel insight.

- **No evaluation of non-transformer self-supervised models**: CNN-based self-supervised methods like MoCo, BYOL, or SimCLR—trained with similar global-invariance objectives—are not tested, leaving a gap in understanding whether attention is necessary for high CSS.
- **Position encoding interpretation not fully addressed**: The task design (patch-based permutation) inherently relies on models using positional information. Those synthesized patches are aligned with the patches of most transformer-based models use (only bigger). So, in effect, transformer models have an advantage over CNN models and transformer models with proper augmentation in training to avoid reliance on the position of the patches should work better on CSS.

---

> ### Author Rebuttal · Authors · 2025-07-30
>
> Thank you for your thoughtful and constructive review! We appreciate your recognition that this is a “creative way to [...]  isolate configural shape sensitivity, controlling for texture completely”. Below, we address specific comments.
>
> ---
>
> > **“CSS is strongly correlated with other shape‑sensitive metrics, so the insight feels incremental.” and “What specific types of failure modes or model differences would CSS detect that previous methods would miss?”**
>
> We respectfully disagree with the reviewer that the correlation with other shape metrics makes our contribution incremental. The shape-vs-texture bias measure is relative and thus can give a false sense of successful shape sensitivity. For example, shape-vs-texture bias can rise simply by suppressing texture (e.g., as in Stylised, Adversarial, Top-k CNNs), while their CSS hardly changes (Fig. 2C). In other words they are not more sensitive to holistic shape (which the CSS measure tells you clearly and directly). Moreover, CSS explains variance on four downstream, shape-dependent benchmarks far better than shape-vs-texture bias (Sec. 9, Williams test), revealing failure modes that earlier diagnostics would have missed.
> In other words, we argue that CSS is a more targeted assay of the underlying construct of interest, which is about how models leverage holistic shape processing (beyond local cues). We will clarify this measure's advantage over existing ones in the Camera-Ready version.
>
> ---
>
> > **“While MAEs are included in the CSS benchmark and perform moderately, the paper does not explore why they underperform compared to models like DINOv2.”**
>
> You are correct and we agree that the moderate CSS performance of MAEs merits further explanation (with the page limit we had to make some choice). To address your concern we ran the representational similarity analysis on Hiera-MAEs and two other masked-image-models in our model suite - BEiT (token prediction) and BEiT-v2 (token prediction with CLIP-distilled targets), which share MAE’s mask-and-reconstruct recipe and differ in the semantic level of the prediction target - target either being 8096-D code-book indices from a vector-quantised VAE that was itself trained (without labels) to reconstruct pixels in BEiT or soft probability vectors distilled from a CLIP vision encoder. We find that only BEiT-v2 develops the mid-layer switch from part-based to category-based representation that we had observed in other high-CSS models (see Fig. 4B). The evidence suggests that just masked modeling (both, pixel or latent level reconstruction) is not sufficient in a purely self-supervised setting, instead DINO’s multi-view contrastive loss that enforces global consistency across large spatial crops appears to be critical to unclocking this configural capacity. When such global pressure is absent, comparable CSS can still be reached by injecting language-aligned targets, as seen in EVA-CLIP and BEiT-v2. Thus, not all self-supervision is equal.
>
> We did not perform attention ablation studies on these MAEs, because probing them with the same radius-mask ablation used for global-attention ViTs is technically challenging without substantial redesign: Hiera-MAE starts with a 7 × 7 conv stride-4, then uses stage-wise local windows that shrink the receptive grid (56 → 28 → 14 → 7). A fixed “mask radius = k tokens” therefore corresponds to four different pixel extents across stages, making a layer-wise comparison impossible without redefining distance at each stage. In stages 0–2 each head already attends only within 8 × 8 or 4 × 4 windows, additional masking outside removes information the model never sees, probing a different phenomenon than our global-attention test. For clarity, we limited Fig. 4 to uniform-grid ViTs (DINO-B/14, ViT-B/16).
>
> We appreciate this suggestion, which helped clarify which self-supervised objectives drive configural sensitivity. We will include this point in the camera-ready version.
>
> ---
>
> > **“High CSS might just reflect heavy spatial augmentations like random cropping in DINOv2.”**
>
> Fair point. We think heavy spatial augmentations alone does not explain high CSS.  Our model suite contains models trained with heavy augmentations like RandAugment (Cubuk et al., 2019), Mixup (Zhang et al., 2018) and Cutmix (Yun et al., 2019):
>
> 1. Multiple variants of ResNets from Wightman et al., 2021. These models are trained with the A1 augmentation recipe with RandomResizedCrop (scale 0.08–1.0), RandAugment, Mixup/CutMix, color-jitter, and flips.
>
> 2. ViT-S/16 AugReg and ViT-B/16 AugReg from Steiner et al., 2021 which applies RandomResizedCrop + RandAugment + Mixup augmentations.
>
> These augmentations however are still fed to the network independently under a cross-entropy loss, no link is made between different views of the same object. The CSS scores for these models reach up to 36.11%. DINO, by contrast, pairs two spatially distant crops of the same image (local and global view) and forces their embeddings to coincide, so the model is directly rewarded for integrating long-range information that spans the global shape of the object in the image. So rather than random cropping which is also present in the DINO framework, we think that the high CSS scores for DINO models can be attributed to its objective that enforces a global-consistency constraint.
>
> ---
>
> > **“Transformer models may benefit unfairly from explicit positional encodings matched to the anagram grid.” and “Have you compared models with different position encodings?”**
>
> Really good point, and we also thought so! We evaluated five new additional variants of the ViT-B/16 architecture from timm, each with a different positional encoding scheme. For context, the standard supervised ViT-B/16 from our paper (which uses learned absolute positional embeddings) has a CSS of 23.61%.
>
> New Experimental Results:
>
> - ViT with Relative Positional Embeddings: 38.89% CSS
> - ViT with RoPE (Rotary Position Embedding): 51.38% CSS
> - ViT with RoPE (Rotary Position Embedding) + Absolute PE: 51.38% CSS
> - ViT with Mixed RoPE: 48.61% CSS
> - ViT with Mixed RoPE + Absolute PE: 52.77% CSS
>
> The stronger performance of models with relative position encodings (like RoPE) demonstrates that CSS does reward models for processing relational spatial structure. We will add this new analysis to the appendix to provide a more complete picture of the role of positional encodings. Thank you for this valuable suggestion.
>
> ---
>
> > **“Can you clarify whether the representational similarity analyses provide insight beyond the expected hierarchical abstraction of deep networks?"**
>
> We agree a simple transition from low-level to task-driven features is expected in deep networks. However, our analysis provides insight beyond this as we found qualitative difference in ‘how’ models make this transition, specifically two distinct patterns:
>
> 1. The "Stuck-on-Local" Pattern: In low-CSS models, the representations remain dominated by local patch similarity across all layers. For the anagram pair (same patches, different objects), the final representations remain highly similar: the model cannot pull them apart. These models fail to make the full transition to a global representation when confronted with matching local evidence.
>
> 2. The "Configural Flip" Pattern: In high-CSS models, we see a non-trivial "representational flip". While early layers are sensitive to the shared patches, the intermediate layers actively work to overcome this. By the final layers, the representations of the two different objects in an anagram pair are just as dissimilar as if they were built from entirely different patches. The model’s representation becomes fully determined by the global configuration, successfully overriding the misleading matched local cues.
>
> Therefore, the insight is not simply that a hierarchy exists. The novel insight is the existence of this "configural flip" as a hallmark of high-performing models and its absence in weaker ones. These high-CSS models are not just going from a local to a task-driven representation, but instead, have an intermediate transition of local enrichment due to long-range contextual interaction which ends up reflecting as the “configural flip” signature. This analysis provides a way to distinguish these models from those whose final output is still contaminated by local feature similarity.
>
> ---
>
> > **“Can you evaluate CNN-based self-supervised models such as MoCo, SimCLR, BYOL, or SwAV to determine whether high CSS requires attention, or if it can arise from training objectives and augmentations alone?”**
>
> We could not include MoCo, SimCLR, BYOL, or SwAV, as pretrained versions with 1000-way classification heads were not available. However, our suite includes SSL-ResNet-50 and SWSL-ResNet-50, which offer partial coverage. We agree this is a valuable direction and plan to test it systematically by holding architecture and data constant while varying the self-supervised objective.
>
> ---
>
> > **“Does the 64×64 patch size of the anagrams mean CSS mostly measures layout rather than sub‑patch integration?”**
>
> Of course CSS requires both subpatch processing and global “layout” sensitivity, but it specifically targets the latter. Since both images in an anagram pair share identical parts, models must rely on configural information to succeed. This is why BagNets, which lack long-range integration, perform at chance.
>
> ---
>
> >  **“Did you try training or fine-tuning DINOv2 (or a similar ViT) without spatial augmentations [...] to assess whether positional robustness is necessary for high CSS?”**
>
> We did not retrain DINOv2 without multi-crop augmentations, as this would require full pretraining from scratch on large-scale data, which is beyond the scope of this paper. Our goal is to evaluate emergent properties of existing models. That said, we agree this is an important question, and a controlled comparison isolating the role of augmentations would be a valuable direction for future work.

---

> > ### Comment · Reviewer_3trt · 2025-08-03
> >
> > Thank you for taking the time to read through the comments and concerns. I appreciate the experiments added regarding the positing encoding. The mechanistic insight from CSS (beyond other shape-sensitive metrics offer) needs more experiments (re: Unclear advantage of the mechanistic analyses in my original comment). However, I maintain the score which is still recommending for acceptance.

---

> > > ### Author Response · Authors · 2025-08-04
> > > **Thanks**
> > >
> > > Thanks again for the follow-up comment.
> > >
> > > Overall, we found your review to be a very good one, it really made us think more deeply about several aspects of the work (e.g., positional encoding, MAEs, loss ablations...). We agree that the mechanistic analyses could have gone further, though we were already at the edge of the 10-page limit and had to make tradeoffs given how many results we included.
> > >
> > > Still, your comments helped us refine both the framing and the experiments, and we really appreciate the time and care you put into the review.

---

### Official Review · Reviewer_L13o · 2025-07-02

**Clarity:** 4
**Significance:** 4
**Originality:** 4
**Rating:** 5
**Confidence:** 4

**Summary:**

This paper defines a measure of image "shape" sensitivity present in trained deep neural networks by using images pairs that are "visual anagrams" of each other.  The method to generate the visual anagrams comes from a previous paper and uses a diffusion model.  The result is a set of image pairs, where each image is a permutation of patches of the other, so texture is locally preserved but the images have different global shape and therefore are associated with different Imagenet categories.  They then construct a score (Configural Shape Score, CSS) to measure the sensitivity of the deep network classification to the texture-preserving permutation by calculating the fraction of times both images are classified correctly, meaning they are both mapped to one of a set of imagenet categories (manually?) set to be correct.  The goal of CSS is to measure "an absolute score of configural shape" without simply suppressing texture, and the authors claim that this score measures a notion of "genuine" shape inference.  They then show that the CSS score varies over the modern spectrum of vision models, with vision transformers generally having higher scores.  They then perform a variety of experiments that study how the CSS score is correlated with the emergence of long-range interactions between patches in transformers, how the representational similarity of image anagrams is inversely related to the representational similarity of different images in the same category as a function of model layer in some models but not others, and how the CSS score is correlated with a selection of other vision model benchmarks.

**Questions:**

* There seems to be a relatively small number of image categories generated.  What is the limitation here?  Why only 9 categories of visual anagrams?  How do these results depend on this?
* You keep using the world "multiset" to describe the set of patches that the visual anagrams share.  How is this a multiset?  Can patches occur more than once in the set?
* You say that this work offers "actionable design principles for future vision systems".  Can you be more explicit about what these design principles are?
* Are rotations of the patches allowed when generating the image anagrams?  This is suggested by figure 1 but I don't see it mentioned anywhere.
* I wonder whether the low performance of some models is because the visual anagrams are out of the training distribution. In particular, the texture cues (e.g. asking for "black paint texture") may be different than images in the training set. Can the authors comment on this possibility? Does this introduce limitations in interpreting the results?

**Ethical Concerns:**

["NO or VERY MINOR ethics concerns only"]

**Final Justification:**

All reviewers seem in agreement that this paper is worthwhile to accept. I think the basic idea is a very clever concept and will be of great interest.

**Limitations:**

yes

**Quality:**

4

**Strengths And Weaknesses:**

**Strengths**

* Overall this is a wonderful paper and I would expect it to be a contender for a spotlight talk if the authors can offer some further improvements during the rebuttal period.
* The idea is clever and well-motivated. It is introduced through lucid writing and clear figures.
* The paper makes a clear and convincing point about shape processing emergence in vision models. In addition to the main findings (which I consider to be the plots in Figure 2),
* The subject matter is perfectly fit for ICLR

**Weaknesses**

* In places, the analyses are somewhat anecdotal. It would increase my enthusiasm for the paper if the authors could be more comprehensive and quantitative.
    * There seems to be a relatively small number of image categories (nine) generated. Also how many image pairs were generated in total? I couldn't find an exact number.
    * Figure 3 shows a suggestive comparison of two networks, but are the trends here consistent across the broader range of transformer architectures? What about transformer architectures that are *trained* with smaller context windows (rather than ablated afterwards)?
* All of the visual anagrams seem to be prompted to have "black paint texture". While this controls that the two images have the same texture, it may mean the texture is out-of-distribution. This complicates my intuition and interpretation of the results (see question below).
* (Minor) The authors use the term "representational similarity analysis" to describe their analysis in Figure 4. This is confusing because Representational Similarity Analysis (RSA; [Kriegeskorte et al. 2008](https://www.frontiersin.org/journals/systems-neuroscience/articles/10.3389/neuro.06.004.2008/full)) is a different technique that is closely related to the popular Centered Kernel Alignment method (CKA; [Kornblith et al. 2019](http://proceedings.mlr.press/v97/kornblith19a.html)) as described in [Williams, 2024](https://www.biorxiv.org/content/10.1101/2024.10.23.619871v1.abstract). These well-known methods focus on comparing representations across different neural systems. To avoid confusion, I suggest the authors come up with different language to describe their results in Figure 4... Maybe something like "measuring representational divergence"?

**Minor/typos:**
- line 63: typo "predictive"
- line 141: grammar, possibly supposed to be "we also added versions"
- Figure 3: "cls" undefined
- Figure 5: caption missing "."

---

> ### Author Rebuttal · Authors · 2025-07-30
>
> We thank you for the generous evaluation and sharp comments. We particularly appreciate the recognition that the study is “clever and well‑motivated,” features “lucid writing and clear figures,” and delivers “a wonderful paper” that could be “a contender for a spotlight talk,” pending minor clarifications. Below, we address specific comments.
>
> ---
>
> > **“There seems to be a relatively small number of image categories (nine) generated. Also how many image pairs were generated in total? I couldn't find an exact number.”**
>
> We agree with the reviewer on the small number of anagrams. The benchmark initially contained 72 total visual anagram pairs, where the choice of the specific 9 categories was motivated by Baker & Elder, iScience, 2022.
> We have now expanded the number of total stimuli by 20x, from 72 to 1440 visual‑anagram pairs. We re-evaluated all models on this new, larger dataset. Preliminary results show that the scores are very stable and the relative CSS rankings of the models are preserved, the correlation between the CSS scores on the original and expanded datasets is r = 0.99. We will include the expanded set and report these results in the camera-ready version of the paper!
>
> ---
>
> > **“Figure 3 shows a suggestive comparison of two networks, but are the trends here consistent across the broader range of transformer architectures? What about transformer architectures that are trained with smaller context windows (rather than ablated afterwards)?”**
>
> Fair point. Our original goal was to provide a clear, controlled comparison between a high-CSS and a low-CSS model with similar ImageNet performance, which is why we focused on DINOv2-B/14 and ViT-B/16. However, since then, we expanded this analysis to the remaining DINO models ( DINOv2-S/14,  DINOv2-L/14,  DINOv2-G/14). When applying the same ablation paradigm, the effects in other models were less pronounced, particularly the deeper models.  We hypothesized the single-block ablation may be too subtle. To address this limitation, we designed a cumulative ablation experiment: First, we ablated all blocks up to block ‘n’ to isolate the contribution of the network up to that point and second we ablated all blocks after block ‘n’ to see if there is recovery by later layers. This analysis showed that the drop is the highest and most irrecoverable (even for the DINO large and giant variants) in the intermediate layers. We will include this cumulative ablation analysis in the camera ready version. While this set is not the full range of transformer architectures, we hope the expanded analysis helps provide more support for the trends we observed. Thank you for this question.
>
> ---
>
> > **“All of the visual anagrams seem to be prompted to have "black paint texture". While this controls that the two images have the same texture, it may mean the texture is out-of-distribution. This complicates my intuition and interpretation of the results (see question below).”**
>
> Indeed, we intentionally used a homogeneous "black paint texture" to control and match the local statistics, as this helps further ensure that any classification difference would stem from the global configuration. However, it’s a fair question to wonder if this unnatural texture is enough out-of-distribution (OOD) that holistic mechanisms observed here do not apply to more ‘natural’ images. To make the concern more concrete, is poor CSS performance actually due to a failure of OOD texture, rather than a failure of configural processing?
>
> There are several results that address this concern. The main evidence comes from single-image accuracy for the anagram images – if the texture itself were the problem, models should fail to classify even individual anagram images. This is not the case. For example, a ResNet-50 classifies single anagram images with 52.7% accuracy (well above the 11% chance rate). Its CSS, which requires correctly identifying both images in a pair, plummets to 16.7%. Similarly Bagnet 33 is at 5% on CSS but at 39% for individual images. This shows the model can extract meaningful features from one image, but fails when it must distinguish two configurations built from the exact same local features.
>
> Further, we also know that poor CSS models are accurately representing the local texture:  the representational similarity analysis shows that these models show high correlations between same-part pairs (Figure 4, green lines), and low similarity between different-part pairs (blue-and orange) indicating they have strong representations of the local details. If all the “black paint textures” were treated the same on account of being out of distribution, none of these results would hold (all correlations would be high, and all three lines would be overlapping).
>
> That said, we agree with you that employing a richer set of matched textures would improve ecological validity, and we will highlight this as an important direction for future work in our discussion.
>
> ---
>
> > **You keep using the world "multiset" to describe the set of patches that the visual anagrams share. How is this a multiset? Can patches occur more than once in the set?**
>
> Apologies for the confusion.  We will change this word to “set” in the places it occurs in the manuscript. (While we were initially trying to use it to clarify the difference between unordered and spatially ordered patches, we realized the sentences in the paper are actually clearer and still accurate when we use the word “set”).
>
> ---
>
> > **"Are rotations of the patches allowed when generating the image…"**
>
> Yes. The transformation involves rotation of the patches. We will update Section 3 in the camera-ready version to explicitly state that the transformation involves both permutation and rotation, which provides the necessary flexibility to form two distinct and recognizable object shapes from the same collection of local parts.
>
> ---
>
> > **You say that this work offers "actionable design principles for future vision systems". Can you be more explicit about what these design principles are?**
>
> For sure, thank you for this question! The core actionable principles derived from our findings are:
>
> 1. Prioritize Training Objectives that Enforce Global Structure. Our results, consistent with prior findings, also show that models trained with standard supervised classification often solve the task using local texture shortcuts. In contrast, self-supervised objectives that require matching global and local views (like in DINOv2) or multimodal objectives that align vision with language (like in EVA-CLIP and SigLIP2) are far more effective at inducing robust holistic processing. Choosing the right objective is critical.
>
> 2. Ensure Architectures Can Perform Long-Range Spatial Integration. Our mechanistic analyses (Sec. 6 & 7) show that the ability to relate distant image patches in intermediate layers is the key computational mechanism behind high CSS performance. The complete failure of architecturally-local models like BagNet reinforces this: without the capacity for long-range communication, models cannot develop configural understanding. It is an open question if post- and pre-training methods that amplify these mechanisms in standard ViTs would unlock such a capacity without needing to train on large orders of data.
>
> 3. Replace shape-vs-texture measure with an explicit shape measure. Our work shows that improving relative shape-vs-texture bias does not guarantee better configural processing. The principle is that these are distinct abilities. Rather than simply discouraging texture use, future work could also directly measure and optimize for absolute configural competence.
>
> We will add a dedicated paragraph in our camera-ready version to clearly outline these principles for future model development.
>
> ---
>
> > **“(Minor) The authors use the term "representational similarity analysis" to describe their analysis in Figure 4. This is confusing [...] To avoid confusion, I suggest the authors come up with different language”**
>
> We see that this terminology is unfamiliar, and we modified the text to more carefully introduce the term. This terminology was originally introduced in the computational neuroscience community (Kriegeskorte, 2008), and is increasingly being adopted in the NeuroAI community (e.g., “Getting aligned on representational alignment”, Sucholutsky 2023), and in ML research focused on similarity of neural network representations (Kornblith et al., 2019).
>
> ---
>
> > **Minor typos**
>
> Thank you so much for pointing these out. We will correct it in the revised draft.

---

> > ### Comment · Reviewer_L13o · 2025-08-03
> >
> > Thank you for your detailed responses. I maintain my high score and hope to see this paper accepted.

---

> > > ### Author Response · Authors · 2025-08-04
> > > **Thanks !**
> > >
> > > Thanks again for the thoughtful and encouraging review!
> > > We really appreciated the time you took and your comments, we genuinely believe they helped make the paper stronger!

---

### Official Review · Reviewer_Bepr · 2025-07-02

**Clarity:** 3
**Significance:** 2
**Originality:** 3
**Rating:** 5
**Confidence:** 4

**Summary:**

This paper presents a new shape processing benchmark that aims to measure whether models focus on local cues or the global configuration of these local cues to recognize objects. They do this by generating pairs of images that share the same local cues but show a different class of object. They use earlier work to generate these visual anagrams where patches in one image are permuted to get the image of another object (e.g., wolf vs. elephant). They define what they call a configural shape score (CSS) using such pairs of images. They test many models ranging from supervised CNNs/ViTs to self-supervised vision and vision + language models and present the following findings.

- DINOv2, SigLIP2, and EVA-CLIP has high CSS, while supervised CNNs and ViT have relatively low CSS.
- Higher object recognition accuracy doesn't necessarily mean higher configural shape score.
- CSS and shape-texture bias are unrelated. Different manipulations that lead to change in shape-texture bias does not affect CSS.
- High CSS models rely more on long range interactions than low CSS models.
  - They find that especially the intermediate layers are important in processing these long range interactions.
- They show the same finding in a different way by looking at similarity between representations where either puzzle pieces or the overall category changes between images.
-  They show that configural shape score also predicts other shape-dependent evals (e.g.,foreground bias, spectral and noise robustness).

**Questions:**

- Why not do the attentional ablation study in section 6 for all or a few more (attenion-based) models? It might be interesting to see if these results hold for other models as well.
- When the images are fed to ViT based models, are you aligning the ViT grid with the anagram's grid? What happens if it is not aligned and one token in ViT actually straddles multiple patches in the anagram? I don't think this will make much of a difference but I was just curious.
- line 298, typo: 2) -> 1)

**Ethical Concerns:**

["NO or VERY MINOR ethics concerns only"]

**Final Justification:**

I think this is a pretty interesting strong paper that deserves to be accepted to the conference.

**Paper Formatting Concerns:**

No concerns

**Quality:**

3

**Strengths And Weaknesses:**

Strengths:

- Presents a very interesting and useful metric for configural processing, which is an important aspect of shape processing
- Evaluated on a wide range of models
- Great analysis of how long range interactions are important and where these happen in a model

Weaknesses:

- I think overall the paper is pretty strong so these are only minor weaknesses.
- It would be good to have more examples (currently 72 pairs) in the dataset to get a more robust score.
- More discussion of the results, especially why certain models have higher CSS, would be helpful. Why do the authors think self-supervised models do better than supervised ones for example?
- It would be good to have a full list of all models in the appendix.

---

> ### Author Rebuttal · Authors · 2025-07-30
>
> Thank you for the clear summary, the nice comments and targeted suggestions. We particularly appreciate your acknowledgement that the paper “presents a very interesting metric for configural processing. Below we address each of your points in turn.
>
> ---
>
> > **“It would be good to have more examples (currently 72 pairs) in the dataset to get a more robust score.”**
>
> We agree that a larger dataset is crucial for ensuring the robustness of our findings. In response, we have increased the corpus to 1440 visual‑anagram pairs ( a 20x increase in dataset size). We re-evaluated all models on this new, larger dataset. Preliminary results show that the scores for models are very stable and the relative CSS rankings of the models are preserved, the correlation between the CSS scores on the original and expanded datasets is r = 0.99. We will include the expanded set and report these results in the camera-ready version of the paper!
>
> ---
>
> > **“Why not do the attentional ablation study in section 6 for all or a few more (attenion-based) models? It might be interesting to see if these results hold for other models as well.”**
>
> We now expand this analysis to the remaining DINO models ( DINOv2-S/14,  DINOv2-L/14,  DINOv2-G/14).  When applying the same ablation paradigm, the effects in other models were less pronounced, particularly the deeper models.  We hypothesized the single-block ablation may be too subtle. To address this limitation, we designed a cumulative ablation experiment: First, we ablated all blocks up to block ‘n’ to isolate the contribution of the network up to that point and second we ablated all blocks after block ‘n’ to see if there is recovery by later layers. This analysis showed that, across all DINO models tested, the drop is the highest and most irrecoverable (even for the DINO large and giant variants) in the intermediate layers. We will include this cumulative ablation analysis in the camera ready version. Thank you for this suggestion.
>
> ---
>
> > **“More discussion of the results, especially why certain models have higher CSS, would be helpful. Why do the authors think self-supervised models do better than supervised ones for example?”**
>
> We were also genuinely wondering about this and we propose that two main factors contribute to high-CSS scores: The first is an objective that includes some form of global/local contrastive loss. For example the high-CSS DinoV2 models include a direct contrastive loss between global views (large crops), and local views (small crops) from the same image. Other high CSS models such as CLIP, SigLIP, SigLIP2, Beit2, Eva-CLIP are directly or indirectly language-aligned. For instance, EVA-CLIP uses masked image modeling but is distilled from a CLIP-trained vision encoder while BEiT-2 is trained to predict masked discrete visual tokens from a semantic CLIP tokenizer. To this end, language, either through direct contrastive learning or via knowledge distillation, provides a “global view” of the image. The pressure for local representations to align with global representations seems to be a key factor giving rise to high-CSS scores, and the ViT architecture (with global attention) seems particularly well-suited to developing high-CSS.
>
> The second possible factor is training on a large, diverse dataset. For example, it seems that the high CSS models were also trained on larger datasets. So, disentangling what CSS benefits are confirmed by the training loss vs the dataset (or an interaction of the two, where benefits of the training objective work best with global attention and at a certain dataset scale and diversity) is important work to be done. We will expand our discussion of these results and our hypotheses about why self-supervised models seem to do better than supervised ones in the camera ready version.
>
> ---
>
>  > **“It would be good to have a full list of all models in the appendix.”**
>
> Absolutely, we will add a detailed list in the appendix for all the evaluated models.
>
> ---
>
> > **“When the images are fed to ViT based models, are you aligning the ViT grid with the anagram's grid? What happens if it is not aligned and one token in ViT actually straddles multiple patches in the anagram? I don't think this will make much of a difference but I was just curious.”**
>
> The pieces of the anagram are not actually aligned with the VIT grid! In fact, the anagram pieces are actually shaped like puzzle pieces (not perfect squares, as was indicated in the caption–we have fixed this)--this means there is no way to align these anagram pieces within the ViT grid. Further, the anagram patches are from a coarse 4x4 grid of 64x64 px pieces, while the ViT16 patches are 16 x16px patches; thus, there are approximately 4 VIT patches per anagram puzzle piece. (To address worries about “border hacking”, in case new local features are created at the interface between puzzle pieces that could account for the performance instead of holistic processes, we have section 8 using bag net models to rule out this possibility.) Thank you for asking these questions; we will clarify these details.
>
> ---
>
> **Summary:** We think with the 20x expansion of the number of anagrams images, and the extended ablation analyses, the paper will be much improved. Thanks for your feedback.

---

> > ### Comment · Reviewer_Bepr · 2025-08-04
> > **Thanks for the response**
> >
> > I'd like to thank the authors for their detailed response; it addresses all my questions. I think the paper is pretty strong and I hope it gets accepted!

---

### Decision · Program_Chairs · 2025-09-17

**Decision:**

Accept (poster)

**Comment:**

This paper introduces the Configural Shape Score (CSS), a novel benchmark using visual anagram pairs to assess holistic shape processing in vision models. It provides compelling evidence that high-performing self-supervised and multimodal transformers achieve stronger configural competence, driven by long-range interactions and mid-layer representational transitions. The authors’ dataset expansion (20× increase) and extended ablation analyses further strengthen the results.

Pros:
* Creative and well-motivated benchmark (visual anagrams) that isolates configural shape sensitivity.
* Extensive evaluation across 80+ models, including CNNs, ViTs, and multimodal systems.
* Strong mechanistic analyses linking CSS to long-range attention and representational “configural flip.”
* CSS correlates with robustness metrics and offers clearer diagnostic power than shape-vs-texture bias.
* Well-written, with thorough responses that add new experiments and clarifications.

Cons:
* Mechanistic insights overlap with known properties (hierarchical processing, long-range integration).
* Limited evaluation of non-transformer self-supervised CNNs (e.g., MoCo, BYOL) leaves some open questions.
* CSS remains correlated with existing shape-sensitive benchmarks, raising questions about incremental value.

All reviewers converge on acceptance, with two recommending spotlight consideration. Despite some concerns about incremental novelty, the benchmark is creative, the analyses are careful, and the expanded dataset makes it a valuable contribution. The work is likely to have significant impact in both neuroscience-inspired ML and vision research.